**Data Availability Statement:** The authors confirm that data from the AUV missions, CTD and FRRf vertical profiles and phytoplankton and

# Contrasting phytoplankton-zooplankton distributions observed through autonomous platforms, *in-situ* optical sensors and discrete sampling

**Glaucia M. Fragoso**[1,2]*, **Emlyn J. Davies**[3], **Trygve O. Fossum**[2,4,5], **Jenny E. Ullgren**[6,7], **Sanna Majaneva**[1], **Nicole Aberle**[1], **Martin Ludvigsen**[2,4], **Geir Johnsen**[1,2]

**1** Trondheim Biological Station, Department of Biology, Norwegian University of Science and Technology (NTNU), Trondheim, Norway, **2** Centre of Autonomous Marine Operations and Systems (AMOS), Norwegian University of Science and Technology (NTNU), Trondheim, Norway, **3** SINTEF Ocean, Climate and Environment, Trondheim, Norway, **4** Department of Marine Technology, Norwegian University of Science and Technology (NTNU), Trondheim, Norway, **5** Skarv Technologies AS, Trondheim, Norway, **6** Runde Forsking AS, Runde, Norway, **7** Institute of Marine Research, Bergen, Norway

* glaucia.m.fragoso@ntnu.no

## Abstract

Plankton distributions are remarkably 'patchy' in the ocean. In this study, we investigated the contrasting phytoplankton-zooplankton distributions in relation to wind mixing events in waters around a biodiversity-rich island (Runde) located off the western coast of Norway. We used adaptive sampling from AUV and shipboard profiles of *in-situ* phytoplankton photo-physiology and particle identification (copepods, fecal pellets and the dinoflagellate *Tripos* spp.) and quantification using optical and imaging sensors. Additionally, traditional seawater and net sampling were collected for nutrient and *in-vitro* chlorophyll *a* concentrations and phytoplankton and meso-zooplankton abundances. Persistent strong wind conditions (~5 days) disrupted the stratification in offshore regions, while stratification and a subsurface chlorophyll maximum (SCM) were observed above the base of the mixed layer depth (MLD ~30 m) in inshore waters. Contrasting phytoplankton and zooplankton abundances were observed between inshore (with the presence of a SCM) and offshore waters (without the presence of a SCM). At the SCM, phytoplankton abundances (*Tripos* spp., the diatom *Proboscia alata* and other flagellates) were half (average of 200 cell L$^{-1}$) of those observed offshore. On the contrary, meso-zooplankton counts were ~6× higher (732 ind m$^{-3}$ for *Calanus* spp.) inshore (where a SCM was observed) compared to offshore areas. In parallel, fecal pellets and ammonium concentrations were high (>1000 ind m$^{-3}$ for the upper 20 m) at the SCM, suggesting that the shallow mixed layer might have increased encounter rates and promoted strong grazing pressure. Low nutrient concentrations (< 1μM for nitrate) were found below the MLD (60 m) in offshore waters, suggesting that mixing and nutrient availability likely boosted phytoplankton abundances. The size of the absorption cross-section ($\sigma_{PII}'$) and yield of photosystem II photochemistry under ambient light ($\varphi_{PII}'$) changed according to depth, while the depth-related electron flow ($J_{PII}$) was similar between regions, suggesting a high degree of community plasticity to changes in the light regime. Our results

zooplankton counts are fully available in the
Supporting Information files.

**Funding:** The authors received funding from the
MoniTARE project (Norwegian Research Council
(NRC), https://www.forskningsradet.no/en/, Project
315514) and the Center of Excellence for
Autonomous Marine Operation and Systems
(AMOS) at NTNU (NRC, Project 223254). The
funders had no role in study design, data collection
and analysis, decision to publish, or preparation of
the manuscript.

**Competing interests:** The authors have declared
that no competing interests exist.

emphasize the importance of using multiple instrumentation, in addition to traditional seawater and net sampling for a holistic understanding of plankton distributions.

## Introduction

Plankton distributions are remarkably heterogeneous, or 'patchy', in the ocean [1]. Plankton patchiness spans many orders of magnitude in time (e.g. biomass fluctuations during high versus low productive seasons) and in space, ranging from micro- (few $m$ to $<1\ km$), meso- (few $km$) to global scales (100-1000s $km$) [2, 3]. At large scales, plankton patchiness can alter ecosystem functioning by sustaining productivity and biodiversity [4, 5], affecting trophic transfer efficiency and promoting ecosystem stability [6].

A particularly interesting form of plankton patchiness is when aggregation occurs within a certain water layer, especially in subsurface waters. The reasons for these aggregations are increased growth rates, vertical migration and/or increased chlorophyll $a$ concentration [$Chl\ a$] per cell [7]. Accumulation of $Chl\ a$ within a subsurface chlorophyll maximum (SCM), is nearly a ubiquitous phenomenon observed in regions with a strong vertical density gradient, including seasonally-stratified, high-latitudinal coastal waters [7, 8]. The vertical accumulation of $Chl\ a$ occurs often within the nutricline, where nutrient concentrations sharply increase with depth, at the same time where access to light becomes limiting for the phytoplankton [9].

The interaction between physical and biological factors has been suggested to control the development of SCMs in the oceans. Physical processes, such as vertical shear, are one of the mechanisms behind the formation of thin horizontal layers ($<1$ m in thickness) of $Chl\ a$, where unicellular plankton are trapped in the nutricline due to constant shear [10]. On the other hand, vertical migration, including controlled buoyancy and convergent swimming, has been assumed to accentuate SCM and to be a strategy for the phytoplankton to overcome nutrient and light limitation [11]. Photo-acclimation can be detected as an increase or decrease of intracellular $Chl\ a$ concentrations (and other light harvesting pigments or photoprotective carotenoids). Increased $Chl\ a$ concentrations for a better harvest of photons at deep, low-light waters have been considered an explanation for SCM across ocean regions [9]. Top-down controls, such as light-dependent grazing of microzooplankton, have been recently suggested to drive SCM formation and biomass accumulation of phytoplankton biomass at depth [12]. Although SCM has been extensively investigated through observational and modelling approaches, a study combining simultaneous measurements of *in-situ* plankton abundance estimations, phytoplankton photo-acclimation and proxies of top-down control (high fecal pellet concentrations) have not yet been carried out.

In this paper, we investigate the bio-physical factors involved in the presence/absence of a SCM in dynamic coastal waters around a biodiversity-rich island (Runde) located off the western coast of Norway. The aim of this research is to investigate the potential processes, such as nutrient and light availability (bottom-up) as well as zooplankton grazing (top-down) in the development of SCMs. For this, we compared two areas with distinct hydrography—offshore–with no SCM formed and subjected to strong wind mixing (50 m mixed layer depth—MLD) versus inshore—more stratified (30 m MLD)–and with a SCM. We used adaptive sampling from AUVs to track tri-dimensional $Chl\ a$ distributions within the upper 60–100 m. In parallel, we analyzed the phytoplankton photosynthetic responses to light intensity, varying with depth and we applied *in-situ* imaging to identify particles such as zooplankton (copepods) and their grazing products (e.g. fecal pellets), as well as the abundance of the dinoflagellate *Tripos*

spp. Finally, we used discrete traditional sampling of nutrients, pigments and the main phytoplankton and zooplankton groups to validate and to complement the data obtained from *in-situ* sensors. The combination of these parameters allowed us to observed contrasting abundances of phytoplankton and zooplankton in a sub-mesoscale space (4 km apart) and to discuss the potential bio-physical factors related to the development of SCMs. The novelty of this study is the combination of multiple instrumentation (AUV and *in-situ* vertical profiles of optical and imaging sensors) and traditional water and net sampling to understand the factors involved in phytoplankton distributions and the presence/absence of patchiness. Our results shed light on the dynamic characteristics of subsurface patchiness in coastal regions exposed to episodic strong wind events and the bio-physical interactions regulating plankton distributions.

## Materials and methods

### Ethics statement

No specific permits were required for the field study as the collection site is not a National Park. This field study did not evolve endangered or protected species.

### Study area

Runde Island is located in Møre og Romsdal county at the western coast of Norway (62.4006 N, 5.6242 E). It is the southernmost, relatively large seabird island along the Norwegian coast. This island is located in a region characterized by a narrow continental shelf with irregular bathymetry and complex hydrography [13, 14]. The Norwegian Coastal Current (NCC) and the Norwegian Atlantic Current (NAC) are the two major water masses around Runde Island [15]. The layering of the two water masses (nutrient-poor NCC on top of the nutrient-rich NAC) results in strong stratification in summer [16], making this a potential area for the formation of SCM. Episodic strong winds can, however, temporarily disrupt this stratification and promote deep water mixing ($> 50$m) [16].

Over the last decades a drastic reduction of seabirds and their reproduction along the Norwegian coast, including at Runde Island, was evident, possibly due to shortage of food (fish) and other natural and human-induced stressors, including competition for food with fisheries, bycatch, and increasing sea surface temperature (SST) [17, 18]. Specifically in Runde, about 75% of Black-Legged Kittiwake *Rissa tridactyla* population declined since the 1980s due to the stock fluctuations (including several collapses) of herring, their essential food source [19, 20]. In the case of the Atlantic puffin *Fratercula arctica*, a more drastic decline has been observed in other regions of the coast of Norway (e.g. Røst, northern Norway) due to the collapse of the spring spawning herring stock in late 1960s, whereas in Runde, a decline of 7% has been observed from 2003–2013 [21]. Because of the emergent decline of seabird populations in Runde Island and the influence of plankton on the biomass on organisms from upper trophic levels [22], it becomes crucial to investigate phytoplankton and zooplankton interactions within SCM in this area.

### Sampling

Sampling around Runde Island occurred between 19th and 20th June 2017 and was performed with an Autonomous Underwater Vehicle (AUV, sensors detailed below) and from shipboard deployments on board of RV *Gunnerus* at five different stations (Fig 1). At each station, vertical *in-situ* profiles of conductivity (salinity), temperature, phytoplankton physiological parameters and particle distributions were conducted, in addition to discrete seawater and net

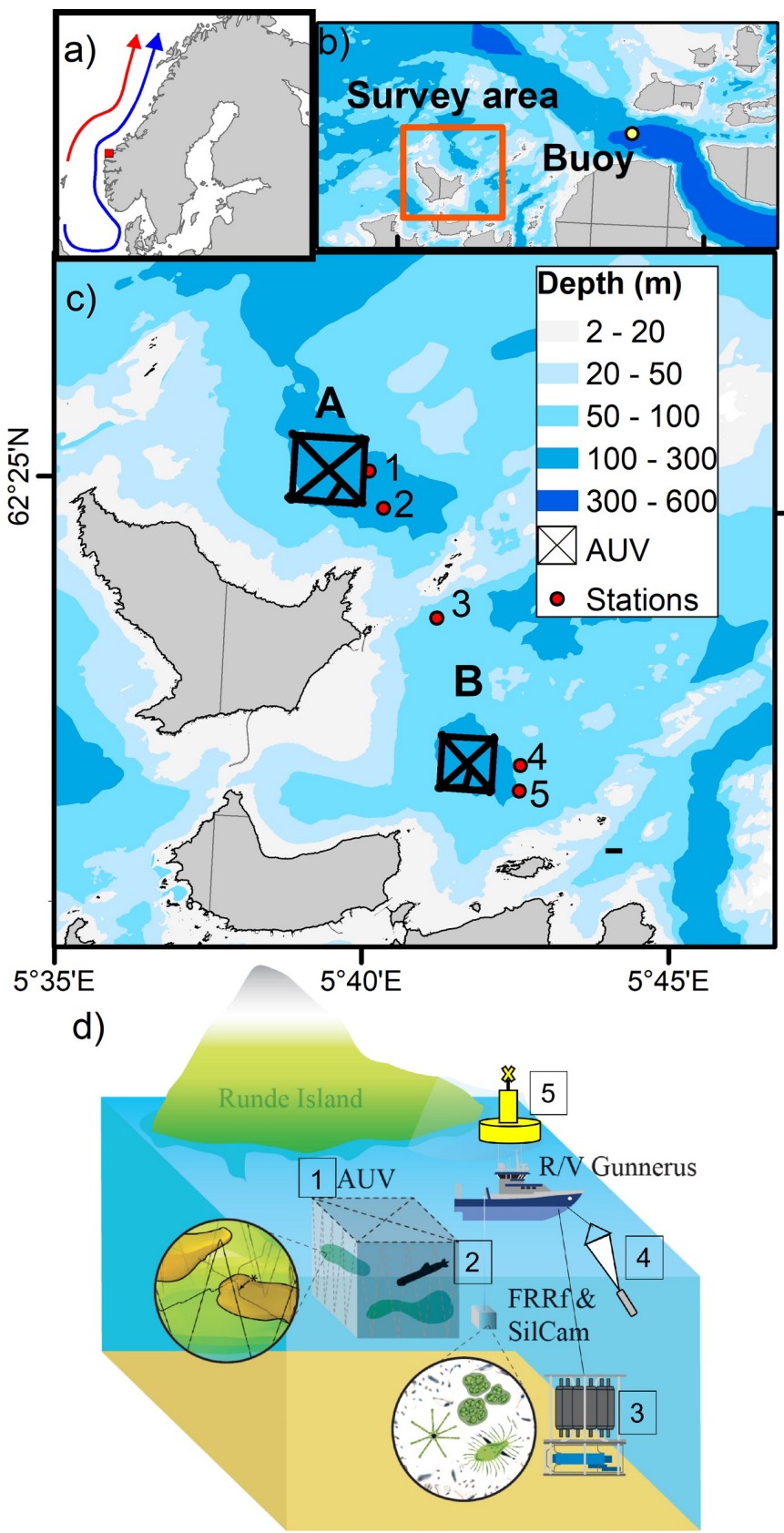

**Fig 1. Study area.** Map showing the location of a) main currents: the Norwegian Coastal Current (blue arrow) and the Norwegian Atlantic Current (red arrow) and Runde Island in the western coast of Norway (red square). b) Map showing the location of the mooring buoy and the survey area. c) Detailed map showing the area where the AUV missions were conducted (box A and B), and stations where shipboard deployments of *in-situ* sensors and discrete seawater and net samples were collected. d) Scheme showing several sampling approaches: 1) AUV surveys, 2) *in-situ* profile of the Silcam for particle imaging and fluorometer for photo-physiological measurements, 3) CTD casts and discrete seawater sampling, 4) zooplankton net tows and 5) time-series of temperature, salinity and wind speed retrieved from the mooring buoy (figure was modified from Fossum et al. (2019)).

sampling (see details below) (Fig 1D). Seawater and net sampling were conducted at approximate time, shipboard measurement occurred on the same day and close in time when the AUV survey was performed for stations 4 and 5 (Table 1). For stations 1–3, shipboard sampling occurred a day before the AUV surveys.

## Buoy data

A moored buoy for meteorological and oceanographic measurements was deployed at Breisundet (62° 26' 44.4" N, 5° 56' 9.9" E), 15 km east of the survey area. The buoy has instrumentation for measuring wind speed, wave height, current velocity, and seawater temperature and conductivity (used to calculate salinity). Only wind speed, temperature, and salinity data from the period 1st March– 31st July 2017 were presented in this study. The conductivity-temperature (CT) sensors, SBE37-SIP, were placed at 1, 20, 40, and 80 m depth and collect data every 10 seconds.

## AUV sampling

A Light AUV (LAUV, OceanScan Light) platform was used to perform a 3D investigation of *Chl a* fluorescence ($FChla_{AUV}$) in the upper 60–100 m in two specific areas around Runde Island (offshore, area A, and inshore, area B) as represented in Fig 1c. The LAUV was equipped with an Eco PUCK sensor (Wet Labs, Oregon, USA) to measure *in-situ* $FChla_{AUV}$ (mg m$^{-3}$, $\lambda_{ex}$ = 470 nm, $\lambda_{em}$ = 695 nm) and a Seabird Fastcat 49 CTD (sampling rate of 16 Hz) for measurements of temperature, salinity (conductivity) and depth (pressure).

The sampling strategy applied in this study is known as AUV-based adaptive sampling [23]. Adaptive sampling is a type of intelligent sampling where the underwater vehicle autonomously makes decisions based on the environmental and vehicle state changes during a mission [24]. Adaptive sampling was specifically chosen in this study to target the spatial changes of $FChla_{AUV}$ to finally recreate the SCM in a tri-dimensional space. The sampling strategy was initially designed to cover the sides of the survey area (Mode 1) and then, recognize specific

**Table 1. Different observational platforms used in this study, their sampling location and time of collection: 1) AUV missions, 2) shipboard measurements, including vertical *in-situ* profiles of imaging (Silcam) and optical sensors (FRRf), in addition to traditional seawater and net sampling and 3) mooring buoy with sensors equipped for wind speed and conductivity (salinity) and temperature at 1, 20, 40, and 80 m depth.**

| Platform | Region/Stations | Day | Time |
|---|---|---|---|
| AUV survey | A | 20.06.2017 | 12:00–14:00 |
| | B | 20.06.2017 | 10:00–11:30 |
| Shipboard sampling | 1 | 19.06.2017 | 12:00–14:00 |
| | 2 | 19.06.2017 | 14:00–15:00 |
| | 3 | 19.06.2017 | 16:30–17:15 |
| | 4 | 20.06.2017 | 11:30–12:15 |
| | 5 | 20.06.2017 | 13:00–14:00 |
| Mooring buoy | Breisundet | 01.03.2017–31.07.2017 | Binned every 30 min |

areas inside the box that have high $FChla_{AUV}$ using a chlorophyll a tracking algorithm (Mode 2). For more details of adaptive sampling used in this study, see [23].

### *In-situ* profiling sampling

A profiling frame was vertically deployed in the upper 100 m of the water column and at approximately 0.2–0.4 cm s$^{-1}$ from RV *Gunnerus* to obtain information regarding the optical and particle properties varying with depth. For that, the frame was mounted with a 1) CTD (SD204 model, SAIV A/S, Bergen, Norway) for conductivity (salinity) and temperature, 2) two Silhouette Cameras (Silcam) for *in-situ* vertical counts of plankton and particle abundances and 3) a Fast Repetition Rate fluorometer (FRRf) connected with a $E_{PAR}$ sensor (irradiance (*E*), measured from 400–700 nm, Photosynthetic Active Radiance, μmol photons m$^{-2}$ s$^{-1}$) for phytoplankton photo-physiology. Data acquisition rates for the CTD were at 1Hz, the Silcam at 7Hz and the FRRf at 0.3Hz.

Two magnification lenses (×0.5 and ×0.25) were used in the Silcam, which allowed *in-situ* determination of the particle size distribution and concentrations from 28 μm to approximately 4 cm in diameter.

Image processing to return particle size, concentration and type, was performed using PySilCam (https://github.com/SINTEF/pysilcam), which utilizes a Convolutional Neural Network for the particle classification component, and is described in [25, 26]. This analysis trains the neural network from a human-labelled database of Silcam images, with 10% of the dataset removed from training for verification purposes. After training, integrated testing of the PySilCam software asserts a minimum accuracy of 96% positive detections across all classes in the neural network, when analyzing the full labelled dataset. Images from the Silcam system were used to estimate abundance of varying types of material present, such as fecal pellets and copepods as described in [13, 26, 27]. For the dinoflagellate genus *Tripos*, formally known as *Ceratium*, only human verification was performed. The images clearly show the presence of copepods, fecal pellets and *Tripos* spp. (see collage of images in Fig 2). Microscopic analyses of phytoplankton and zooplankton (see Seawater and net sampling and Results section below) was obtained to support and verify the content provided by the Silcam.

Photosynthetic parameters, using variable Chl *a* fluorescence kinetics, was measured with an *in-situ* FastOcean FRRf attached to an underwater downwelling irradiance sensor ($E_{PAR}$, Chelsea Technologies Group Ltd, UK). Because the FRRf was run in an autonomous deployment mode (from a battery pack), a series of three different single turnover (ST) acquisition protocols were set prior deployment. These protocols used different combinations of LED type (blue, green and red), irradiance flux ($E_{LED}$) and number of saturation and relaxation flashlets (see details in S1 Table). The different protocols were used to select the best $R_{\sigma PII}$ value (dimensionless), which indicates the optimum combination of intensity and color of the LED that fully saturates the reaction centers during the first flashlet. This allows proper estimation of the functional absorption cross-section of photosystem II (PSII) photochemistry under ambient light ($\sigma_{PII}$') [28]. The optimum $R_{\sigma PII}$ usually lies between 0.4 and 0.6 and the data from the protocol that had the best $R_{\sigma PII}$ were chosen (in this case protocol A, S1 Table). The *Chl a* fluorescence data from the FRRf ($FChla_{FRRf}$) were processed using the FastPro software (Chelsea Technologies Group Ltd, UK) and provided several parameters in this study (Table 2). Those were: 1) the light-regulated quantum yield of photochemistry in PSII ($\varphi_{PII}$', dimensionless), which is related to the fraction of absorbed photons used for photochemistry; 2) the absorption cross-section for PSII photochemistry ($\sigma_{PII}$', nm$^2$ PSII$^{-1}$), which is the light-adjusted response of the physical size of the antenna and $\varphi_{PII}$' to mitigate over- or under-excitation of PSII [29]; 3) the photochemical flux per PSII ($J_{PII}$, electrons PSII$^{-1}$ s$^{-1}$), which is a product of $\sigma_{PII}$' and

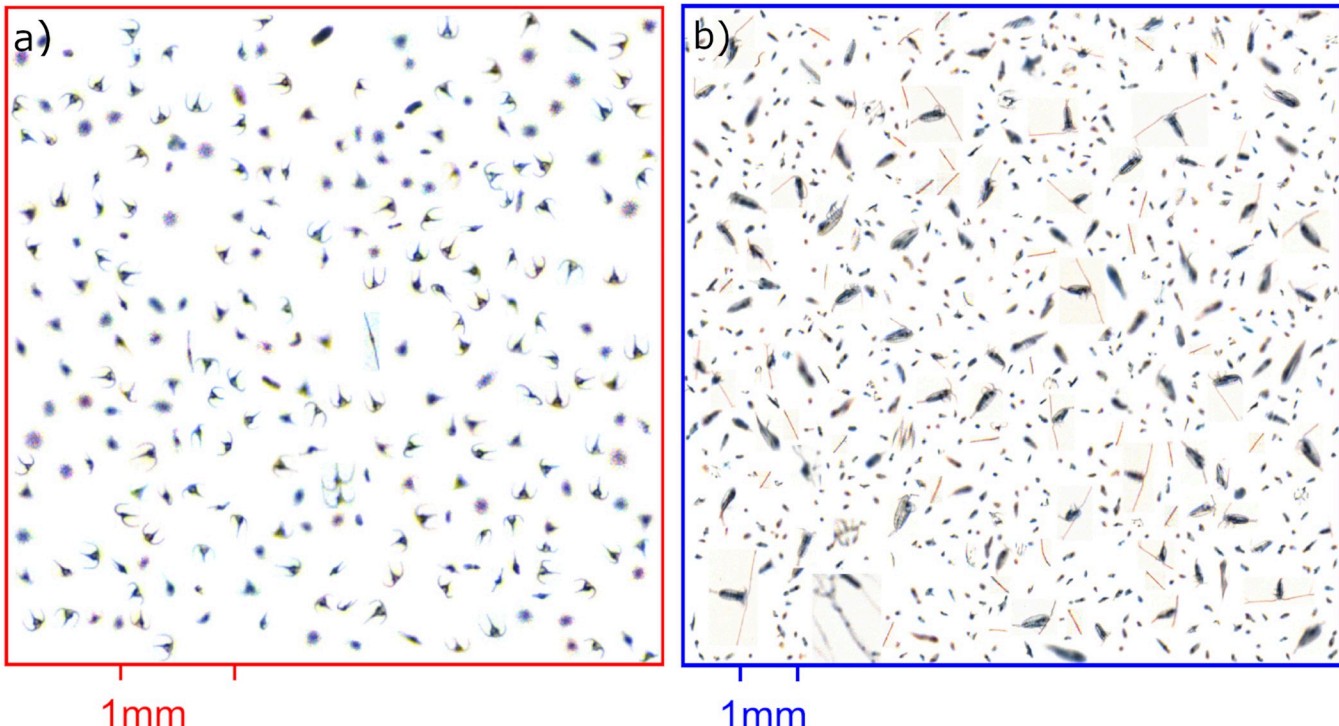

**Fig 2. Silcam images.** Collages of particle images collected from the upper 20 m of a) stations 1 & 2 using the high magnification lens and b) stations 4&5 with a low magnification lens. Note the presence of several species of *Tripos* in a) and copepods and fecal pellets in b).

$E_{PAR}$ and is used to estimate photochemical flux through each open reaction center of PSII; and 4) the proxy that defines the optimum combination of intensity and color of the LED that fully saturates the reaction centers during the first flashlet ($R_{\sigma PII}$, dimensionless) (see parameters definition in Table 2 and review of these parameters in [30]).

**Table 2. List of measured variables, their symbology and unit measurement.**

| Parameter | Description | Unit |
|---|---|---|
| *Chl a* | Chlorophyll a, brackets indicate concentration | mg m$^{-3}$ |
| *Chla$_{in-vitro}$* | *In-vitro* chlorophyll *a* | mg m$^{-3}$ |
| *FChla$_{AUV}$* | Chlorophyll a fluorescence derived from AUV | mg m$^{-3}$ |
| *FChla$_{FRRf}$* | Chlorophyll a fluorescence derived from FRRf | mg m$^{-3}$ |
| *E$_{PAR}$* | Irradiance (*E*), measured in the spectral window 400–700 nm, Photosynthetic Active Radiance (PAR) | μmol photons m$^{-2}$ s$^{-1}$ |
| $\phi_{PII'}$ *or* $F_q'/F_m'$ | PSII photochemistry yield under ambient light | dimensionless |
| *J$_{PII}$* | Photochemical flux through each open reaction center | electrons PSII$^{-1}$ s$^{-1}$ |
| *R$\sigma_{PII}'$* | Optimum combination of intensity and color of the LED that fully saturates the reaction centers during the first flashlet | dimensionless |
| $\sigma_{PII}'$ | Absorption cross-section of PSII photochemistry under ambient light | nm$^2$ PSII$^{-1}$ |
| *NPQ* | Non-photochemical quenching | dimensionless |
| *E$_{LED}$* | Irradiance (photons) from the FRRf measuring LEDs during a flashlet | photons nm$^{-2}$ 100 μs$^{-1}$ |
| *rETR* | Relative PSII electron transport rate | $E_{PAR} \times \phi_{PII'}$ |
| *k$_d$* | Diffuse light attenuation coefficient | m$^{-1}$ |

The diffusive ambient light attenuation coefficient, $k_d$, was calculated as: $k_d = 1/z^* ln (E_0/E_z)$ according to [31], where $E_0$ is the underwater $E_{PAR}$ at surface waters ($< 3$ m) and $E_z$ is the $E_{PAR}$ at depth $z$. Low $k_d$ values suggest low diffuse light attenuation coefficients (clear waters), whilst high $k_d$ indicates more attenuation in the water column due to high particle and colored dissolved organic matter (cDOM) concentrations [32].

## Seawater and plankton net sampling

Seawater samples were collected every 10 m from surface ($<5$ m) down to 60 m using 2.5 L Niskin bottles mounted on the CTD rosette frame and filtered on board of RV *Gunnerus*. For nutrient analyses, water samples were filtered using a 0.8 μm polycarbonate filter and the filtrate was immediately placed in a centrifuge tube and kept frozen at -20˚C. Nutrient concentrations (nitrate + nitrite, silicate, phosphate and ammonium) were analyzed in the laboratory using a continuous flow automated analyzer (CFA, Auto-Analyzer 3, SEAL). For *in-vitro* chlorophyll concentrations determined by fluorometry ($Chla_{in-vitro}$), seawater was filtered (0.25 L– 0.5 L, depending on biomass) onto a Whatman GF/F glass fiber filter. Each filter was double-folded, wrapped in aluminum foil and immediately placed at -20˚C for *a posteriori* analyses in the laboratory. For phytoplankton identification and enumeration, water samples were directly placed into dark amber bottles and immediately fixed with neutral Lugol's iodine solution to a final concentration of ~1%. Lugol samples were stored at room temperature and in the dark for later microscopic analyses in the laboratory.

Taxonomic identification and quantification of phytoplankton (size $>$~4 μm) from Lugol samples were conducted. Aliquots of 50–100 ml from preserved Lugol samples were transferred to Utermöhl chambers and analyzed after 24h using a Leica DM IRB inverted microscope with ×200 and ×400 magnification. Small ciliates and phytoplankton (here defined as $> 4$ μm and $< 30$ μm) were identified as ciliates ($<30$ μm), cryptophytes, chrysophytes, dinoflagellates ($<30$ μm) and other unidentified flagellates and counted on a transect of the chamber at ×400 magnification. Large phytoplankton and ciliates ($>30$ μm) were counted during full examination of the settling chamber at ×200 magnification. Calculations were performed to provide taxa cell counts per L. Phytoplankton were identified to class, genus or species, whenever possible, according to [33, 34].

Oblique plankton net tows were performed for zooplankton abundance estimations on board of RV *Gunnerus* using a WP-3 net (Hydrobios, Ø 113 cm, 500 μm, oblique tows from 20 m depth to surface) equipped with a 2 L cod-end. Samples were preserved with $>70\%$ EtOH and stored at room temperature and in the dark for later microscopic analyses in the laboratory. Taxonomic identification and quantification of zooplankton from EtOH samples were conducted using a Leica (model M205C) microscope. Zooplankton were identified to class, genus or species according to [35] and calculations were made to provide the taxa individual counts per m$^3$.

## Pigment analyses

*Chla$_{in-vitro}$* were determined 4 months after collection. For a fluorometric-based analyses, *Chl a* was extracted in 100% methanol after 2 hours placed on a dark fridge at -10˚C, and determined using the Turner Designs Trilogy fluorometer (model: 7200–000) and the non-acidification method [36].

*In-situ* surface ($<10$ m) *FChla$_{AUV}$* and *FChla$_{FRRf}$* are often spurious. Such measurements are not always representative of phytoplankton biomass concentrations because of a phenomenon named non-photochemical quenching (*NPQ*). *NPQ* is a photo-physiological response of live cells to high light levels, where the excess of energy is dissipated as heat, instead of being

used for photochemistry [37]. Thus, a reduction (quenching) of the $FChla_{AUV}/FChla_{FRRf}$ signal is often induced by high light conditions, particularly at surface waters during daytime hours [38]. To cross-check for the presence or absence of $NPQ$, $Chla_{in\text{-}vitro}$ measurements from discrete water samples (1, 10, 20, 40 and 60 m) were taken to validate the data retrieved from sensors.

### Data analyses

To compare the contrasting horizontal sub-mesoscale (~ 4 km) and vertical distributions of phytoplankton and meso-zooplankton related to the presence/absence of a SCM, data from discrete water and net samples (nutrient concentrations and plankton abundances) were pooled (averaged per depth) as: stations 1 and 2, and stations 3, 4 and 5. Pooling of these stations were based on similar hydrographical structure (temperature, salinity and mixed layer depths from CTD profiles, which influenced the vertical patterns of biochemical factors (nutrient concentrations and phyto- and zooplankton abundances). The AUV profiles from distinct areas (survey box A performed offshore and survey box B—inshore) reinforce the two distinct mixed layer depths (50 m from area A and 30 m from area B, with the presence of a SCM). For the Silcam data (plankton and fecal pellets), counts were pooled for stations (in this case: 1 & 2 and 4 & 5) and binned for the following depths: 0.5–20 m, 20–40 m and 40–60 m.

Boxplots were used to observe the variability of nutrient concentrations and phytoplankton and zooplankton abundances within pooled stations and depths. Local weighted regression (loess in Matlab) was used for fitting a smooth curve in the scatterplots to observe the trends of photosynthetic parameters derived from the FRRf varying with depth, such as of $FChla_{FRRf}$, $k_d$, $\phi_{PII}$', $\sigma_{PII}$' and $J_{PII}$.

## Results

### Hydrographic measurements

Salinity and temperature data collected from the moored buoy at Breisundet varied in depth and time (Fig 3A and 3B). The water column was relatively well mixed until mid-April, where salinity and temperature varied from 32–33.5 and 5–8˚C, respectively, within the upper 80 m (Fig 3A and 3B). From May onwards, the water column, in general, became gradually more stratified (Fig 3A and 3B).

In mid-June, during the cruise period, strong winds due to a local storm were persistent for 4–5 days, which disrupted the stratification typically observed in early summer (Fig 3C and 3D). Shipboard measurement occurred when the strong winds were still present (Fig 3D). Such episodic event allowed for mixing (down to 50 m) of exposed off-shore waters around Runde (near region A), where $FChla_{AUV}$ was evenly mixed from surface to 50 m deep (Fig 4A and 4C). However, in more protected (near the coastline and surrounded by islands) waters (region B, ~4 km from region A), a subsurface (~20–30 m deep) $FChla_{AUV}$ maximum was observed, along with a shallower mixed layer depth (MLD ~30 m) (Fig 4B and 4D).

The vertical structure of the water column from shipboard (CTD rosette) and AUV surveys presented similar pattern (Fig 4E and 4F), even though shipboard CTD profiles occurred a day before (19th June) than the AUV (20th June). A SCM was confirmed inshore in both days (with the AUV data at area "B" on the 20th June and shipboard measurements of $FChla_{FRRf}$ on the 19th June, Fig 8A). At stations 1 & 2, located near region A, the MLD was around 50 m as compared to 30 m observed at stations 3, 4 & 5 (near the region B). Offshore regions (stations 1 & 2 and AUV survey box "A") had a deeper mixing layer (50 m) possible because this area was more exposed to winds. Inshore regions (stations 3–5, AUV box area "B"), which was likely

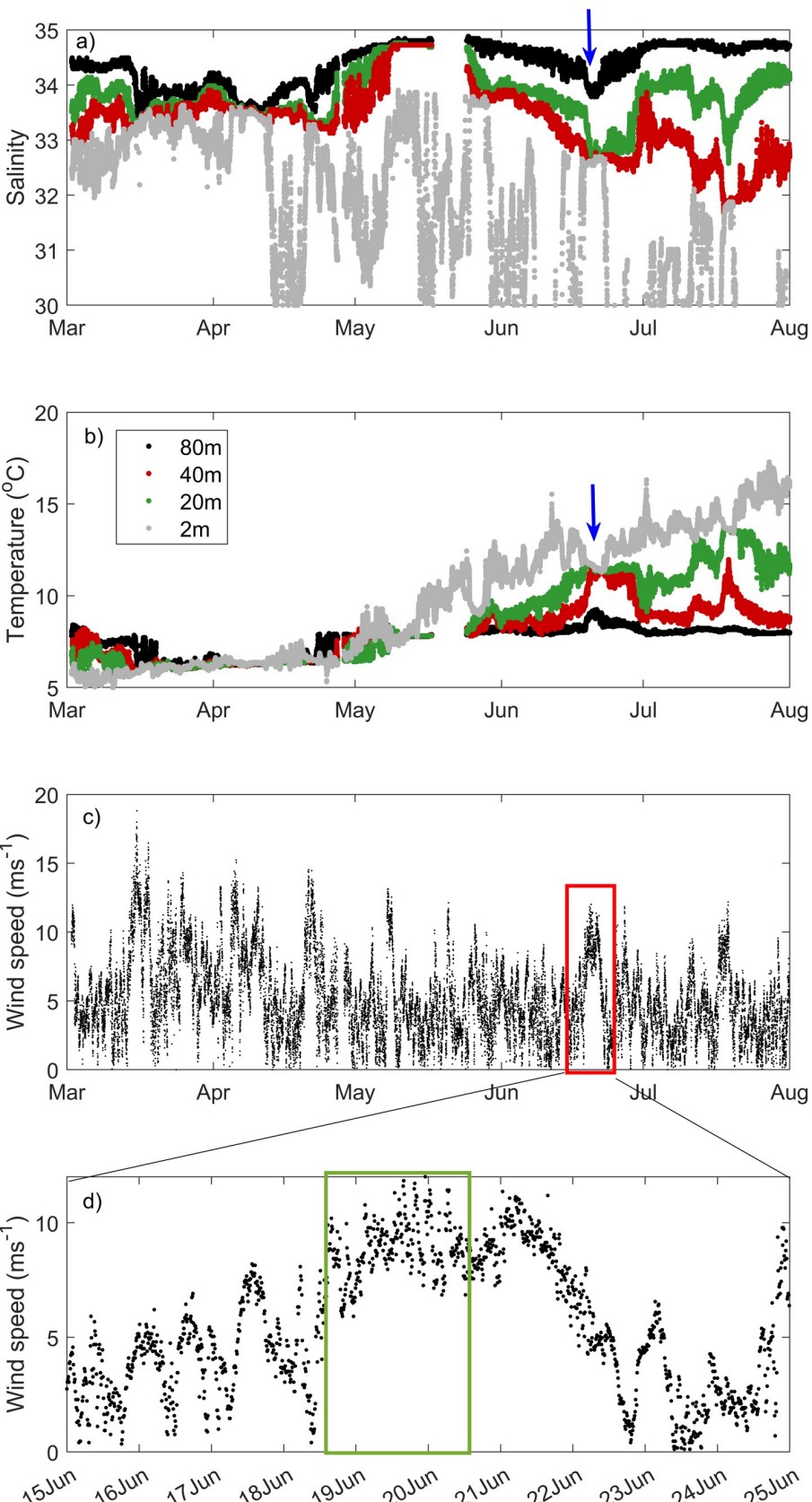

**Fig 3. Buoy data.** Time-series of a) salinity, b) temperature (˚C) from 2, 20, 40 and 80 m and c) wind speed (m s$^{-1}$) collected every 10 minutes from the 1$^{st}$ March until 1$^{st}$ August 2017 and d) zoomed from 15$^{th}$ June– 25$^{th}$ June. Note the period of intense mixing in mid-June (arrows and boxes), which coincides with the time of field sampling: 19$^{th}$ and 20$^{th}$ June (blue arrows in a, b and red and green boxes in c,d).

more protected from winds because it is surrounded by islands, had a shallower mixed layer depth (30 m).

## Nutrient concentrations

The vertical profile of nutrients showed different patterns for the pooled stations (50 m versus 30 m MLD). For $NH_4$ concentrations, stations with shallower MLD (30 m) had generally higher values at all depths than stations with deeper MLD (50 m) (Fig 5A). Concentrations of $NH_4$ were also relatively high at the surface (average of 0.4 μM), being low (<0.4 μM) from 10–40 m and increasing again at 60 m (average of 0.5 μM and 0.9 μM for stations with 50 m and 30 m MLD, respectively) (Fig 5A). Concentrations were low from surface to 40 m for $NO_3$ (< 1 μM), $PO_4$ (< 0.1 μM) and $Si(OH)_4$ (< 1.5 μM) in both pooled stations (Fig 5B–5D). At deeper waters (~ 60 m, particularly at stations with shallower MLD (30 m), the values of these nutrients were from 4–5× greater than the surface values (Fig 5B–5D).

## Phytoplankton and zooplankton enumeration from seawater and net samples

In general, ciliates and phytoplankton, such as *Proboscia* spp. and *Tripos* spp., and other flagellates (cryptophytes, dinoflagellates–other than *Tripos* spp.—and unidentified flagellates) were more abundant at stations with deeper MLD (50 m) (Fig 6A–6D). *Proboscia* spp. and *Tripos* spp., for instance, had concentrations 2–10 × higher (up to 500 cell.L$^{-1}$ in the upper 40 m) at stations with 50 m MLD than stations with 30 m MLD (30 m) (Fig 6B and 6C). At stations with deeper MLD (50 m), phytoplankton and ciliates were evenly distributed from the surface down to 40 m depth, except for *Proboscia* spp. and flagellates other than *Tripos* spp., which had lower abundance at 10 m (Fig 6B and 6D). At stations with 30 m MLD, ciliates peaked at the surface (1 m) and flagellates (included *Tripos* spp. among others) peaked at the surface and 20 m, while *Proboscia* spp. was more abundant at the 1, 20 and 40 m (Fig 6A–6D). The most dominant species of *Tripos* were *T. tripos*, *T. longipes*, *T. fusus*, *T. macroceros* and *T. lineatum* (S2 Table).

In terms of meso-zooplankton taxa, copepods, particularly *Calanus* spp., were the most abundant taxa, followed by Cladocera, Euphasiidae, Gastropoda veliger and Hydrozoa (Fig 6E). When comparing between different sites, zooplankton and phytoplankton abundances had the opposite trends, as zooplankton was observed in higher concentrations at station with a shallow MLD and had lower numbers in stations of deep mixing (Fig 6EE). It was mainly copepods and cladocerans that contributed to the main difference.

## Particle estimations from the Silcam

Images collected from the Silcam showed that particle concentrations and type (*Tripos* spp., fecal pellets and copepods) varied between pooled stations (of a MLD of 50 m and 30 m) and depth (1–20 m, 20–40 m, and 40–60 m) (Fig 7, S1 and S2 Figs). In both sites, particle sizes (from 10–10$^4$ μm equivalent circular diameter) and concentrations (10$^{-5}$–10$^2$ counts/L/μm) showed a negative correlation (S1 and S2 Figs). Small particles were more abundant than large ones, following the typical power-law distributions observed in the oceans and known as the Junge slope (S1 and S2 Figs). A small deviation of the Junge slope from the high magnification

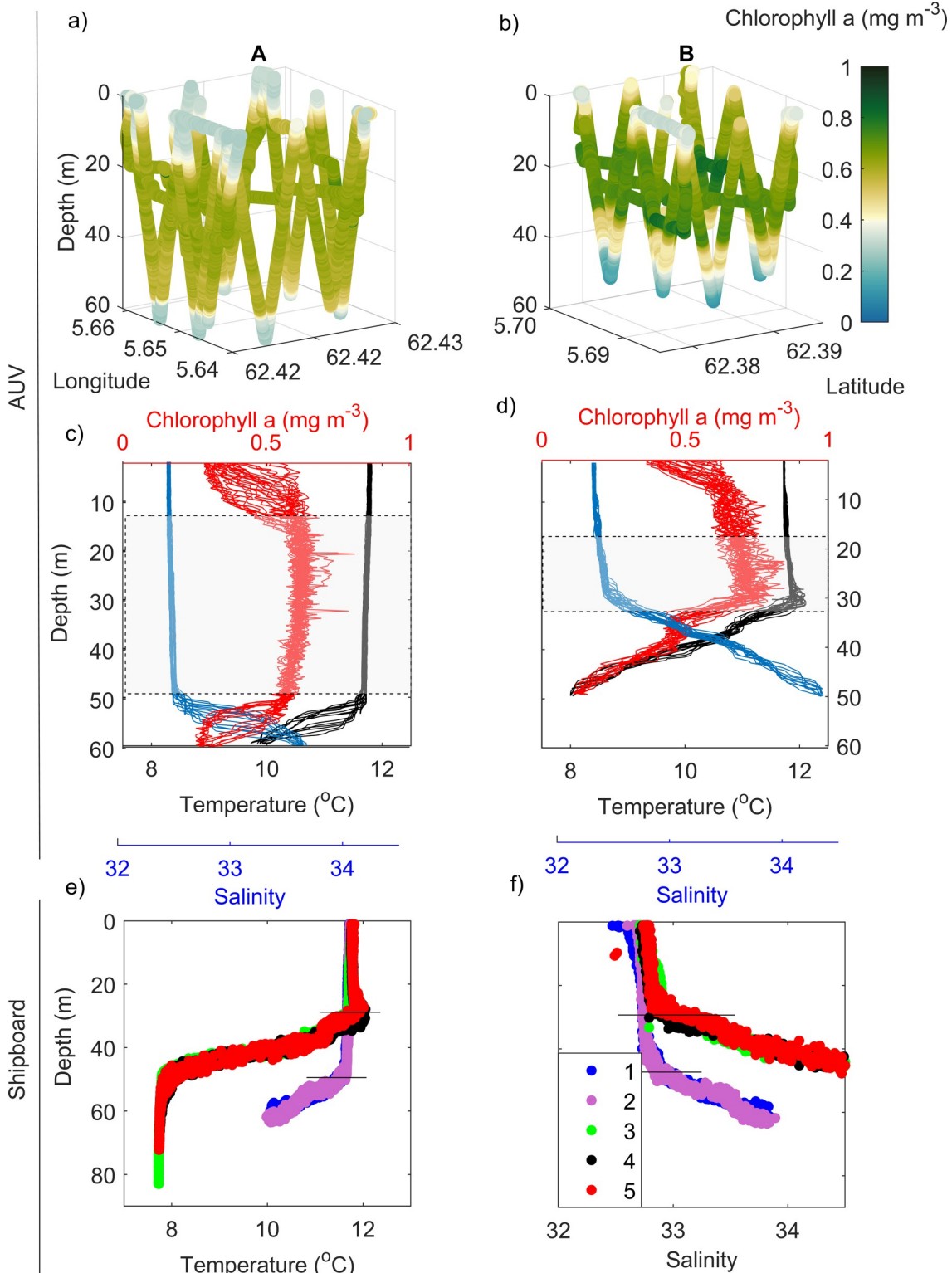

**Fig 4. Vertical profiles.** (a,b) Tri-dimensional *in-situ* profiling of AUV-derived *chlorophyll a* fluorescence ($FChla_{AUV}$, mg m$^{-3}$). c,d) Vertical distributions of $FChla_{AUV}$ (mg m$^{-3}$, red line), temperature (˚C, black line) and salinity (blue line) derived from AUV from the two survey areas (A-left and B-right, see survey areas in Fig 1C). Shipboard vertical profile of e) temperature (˚C) and f) salinity from stations 1–5 with distinct mixed layer depth (30 m at stations 3, 4 and 5 and 50 m at stations 1 and 2). Note the dashed box in c,d showing the highest values of $FChla_{AUV}$ and the black lines in e,f showing the depth of the mixing layer from each station.

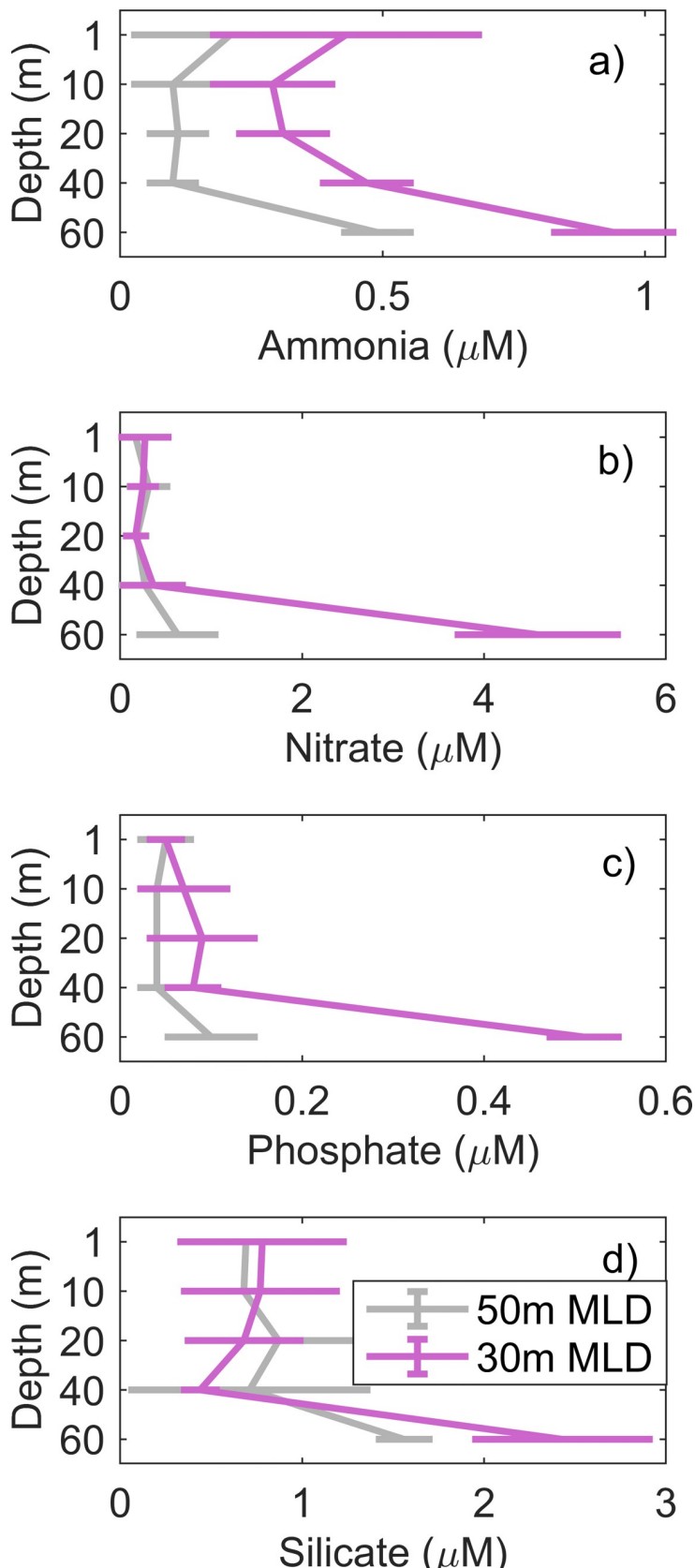

**Fig 5. Nutrient concentrations.** Mean vertical concentrations of a) ammonia, b) nitrate, c) phosphate and d) silicate (μM) from stations where the mixed layer depth was deep (50 m, grey) and shallow (30 m, pink).

lens (×0.5) represents a peak in particle concentrations (from 100–160 μm), which through image analyses, were identified mostly as dinoflagellates *Tripos* spp. (S1 and S2 Figs). For the low magnification lens (0.25×), fecal pellets and copepods represented most particles from 400–600 μm and 800–1200 μm, respectively (S1 and S2 Figs).

For *Tripos* spp., for which images and counts were extracted using a high magnification lens, abundance was ~10× higher (from 600 to 800 cells L$^{-1}$) in waters of deeper (50 m) than shallower mixing (30 m) (Fig 7A). In terms of vertical distributions, *Tripos* spp. abundances were slightly higher at 20–40 m (800 cell L$^{-1}$) in sites with deep mixing layer (50 m) but, in general, values were similar through different depths in shallow mixing waters (30 m) (Fig 7A). Due to technical problems in the low magnification lens in stations of deep mixing, there is only data from abundance of fecal pellets and copepods from stations with shallow mixed layers. For these stations, fecal pellets (> 1000 ind m$^{-3}$) and copepods (average of 200 ind m$^{-3}$) were concentrated in the upper 20 m, decreasing in abundance with depth (500 and 26 ind m$^{-3}$, respectively, from 40–60 m) (Fig 7B and 7C).

## Photosynthetic parameters

Because *in-situ* surface (< 10 m) chlorophyll fluorescence measurements (from the AUV and the FRRf) are often spurious due to *NPQ*, *Chla$_{in-vitro}$* from discrete water samples (1, 10, 20, 40 and 60 m) were taken to validate the data retrieved from *in-situ* sensors. According to the *FChla$_{FRRf}$* vertical profile and *Chl$_{in-vitro}$* measurements, *in-situ* and *in-vitro* chlorophyll measurements from several depths matched up, which confirms that a SCM was present at stations with shallower MLD (30 m, near region B) as compared to stations with deeper MLD (50 m, near region A) (Fig 8A).

The vertical profile of the diffuse light attenuation coefficient ($k_d$) was different between pooled stations. Values of $k_d$ were higher at stations of shallow mixing layer (30 m) compared to deep mixing (50 m), suggesting that the waters were less clear possibly because of high particle concentrations (Fig 8B).

Photochemical parameters, such as φ$_{PII}$' and σ$_{PII}$' varied differently at depth between the pooled stations with distinct MLD (Fig 8C and 8D). At stations with deeper MLD (50 m), φ$_{PII}$' and σ$_{PII}$' increased gradually with depth, with maximum values around 50 m, decreasing sharply with deeper waters (Fig 8C and 8D). On the other hand, at stations with shallower MLD (30 m), φ$_{PII}$' and σ$_{PII}$' increased more steeply and had higher values ~30 m depth (Fig 8C and 8D). Although φ$_{PII}$' and σ$_{PII}$' had different patterns between pooled stations, $J_{PII}$ and the initial slope ($\alpha$) in photosynthesis versus irradiance curve, here represented as relative electron transport rate (*rETR* versus $E_{PAR}$) were similar (Fig 8EE and 8F).

## Discussion

### Patchiness of plankton communities

SCM are typically observed in coastal Norwegian waters during summer [39]. In this study, the waters around Runde showed an increased stratification from spring to summer, consistent with observations found in other regions in the coast of mid-Norway [16]. However, in this study, SCM was only observed in less exposed waters (more inshore, region B), where the mixed layer was shallow (30 m). In more exposed waters (region A), strong winds temporarily disrupted the stratification and [*Chl a*] were well distributed throughout the mixed layer

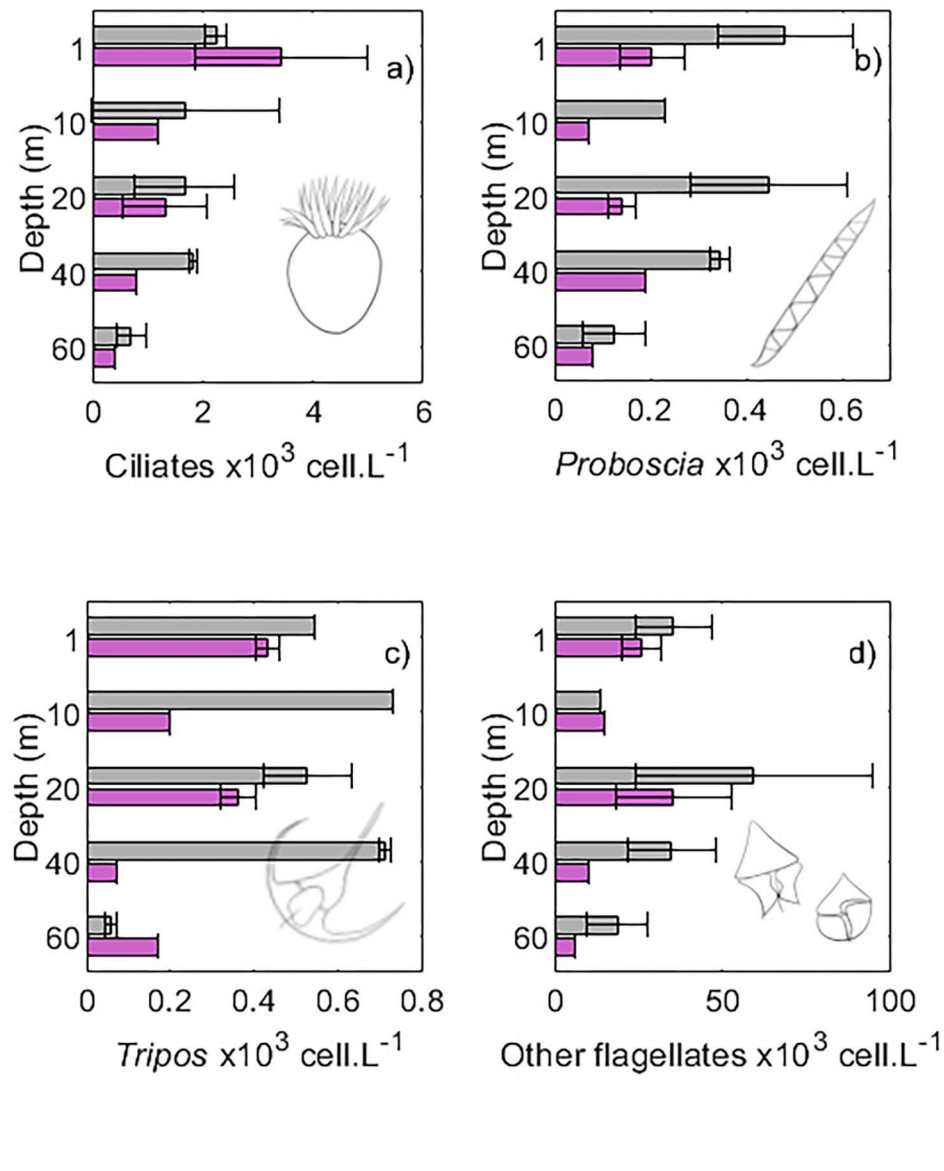

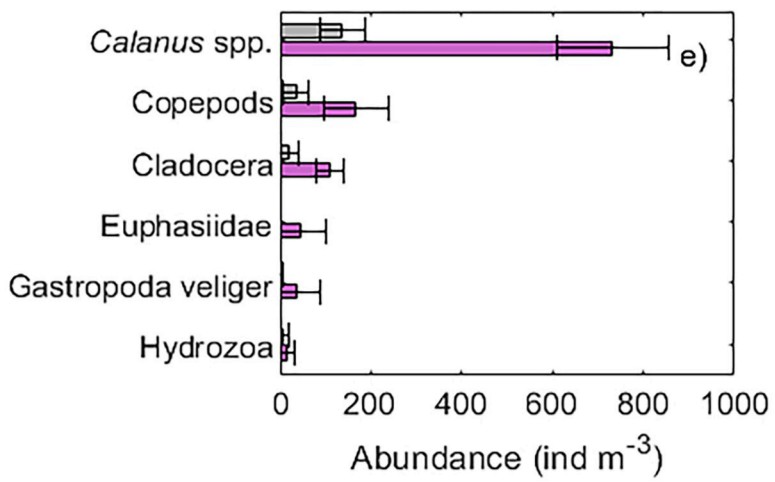

**Fig 6. Ciliates, phytoplankton and meso-zooplankton abundances.** Average abundance (from traditional microscopy) of a) ciliates, b) the diatom *Proboscia* spp. c) the dinoflagellate *Tripos* spp., d) other flagellates and e) meso-zooplankton groups. Colors represent pooled stations from offshore (1 and 2, grey) and inshore (3–5, pink). Other flagellates here refer to cryptophytes, dinoflagellates (other than *Tripos* spp.) and unidentified flagellates (see Materials and Methods).

(upper 50 m). Patchiness in this study was also observed between distinct regions, within the sub-mesoscale (4 km apart) distance. This study confirms the findings from a previous study [39], where the presence of the SCM was strongly determined by the vertical structure of the water column.

Stations with deep mixing (50 m) had higher phytoplankton and ciliate abundances, whilst nutrient drawdown was noticed throughout the upper 60 m, suggesting that access to nutrients from the deep might have boosted primary production and the growth of primary consumers (ciliates). Deep mixing caused, for instance, by strong winds or fronts, has been suggested to sustain SCM in seasonally-stratified shelf seas [40] and in frontal regions of the Fram Strait [41]. This study provides evidence that an episodic wind surge during summer temporarily disrupted the vertical stratification of the water column. Such an event potentially promoted nutrient injections from deep to surface waters, boosting the growth of phytoplankton in stations with deep mixing. Alternative to this, deep mixing might have slowed down predator-prey interactions and the grazing rates, allowing for a strong decoupling between phytoplankton and zooplankton concentrations observed in both regions. Physical disturbances, such as deepening of the mixed layer, has been hypothesized to "dilute" the encounter rates of herbivore grazers and viruses, allowing for the accumulation of phytoplankton cells [42].

Despite the disruption of stratification in more exposed regions, phytoplankton community composition was consistent with those usually found in stratified, thin SCM layers [8, 43, 44]. For instance, the dinoflagellates *Tripos lineatum*, *T. fusus* and the diatom *Proboscia alata* were the predominant phytoplankton species found in subsurface thin layers in seasonally-stratified waters of the Western English Channel [8, 45]. In this study, these three species were found in all stations, although *T. tripos* and *T. longipes* were the dominant *Tripos* species. This could indicate that they could have been present before the storm surge, when layers were stratified for a significant amount of time, but cell division was boosted once nutrients became available through mixing (doubling rate of *T. tripos* is 0.25 d$^{-1}$) [40]. However, rather than rapid cell division, subtle imbalances in predator-prey relations, such as reduced encounter and grazing rates [42], might explain why *Tripos* spp. was abundant in deep mixed waters.

There are many ecological strategies related to the ability of rhizosolenids, including *P. alata*, and the dinoflagellate *Tripos* spp. in thriving within SCMs. *P. alata*, as well as other rhizosolenids might take advantage of low light conditions due to their high aspect ratio [46]. Such a trait allows them to redistribute their chloroplast throughout the cell and, if cells were oriented horizontally, they can increase their light absorption efficiency under deep, low light conditions [46]. Rhizosolenids can also regulate their positioning in the water column through buoyancy [47], and are able to store nutrients when they are abundant and rely on this source, when nutrient input is pending. In the case of *Tripos lineatum*, *T. tripos* and *T. longipes*, these dinoflagellates are primarily photosynthetic but can change to mixotrophic mode to cope with nutrient-poor conditions in highly stratified waters throughout the summer [8]. *Tripos* spp. also have superior swimming ability compared to other dinoflagellates (up to 98 cm h$^{-1}$ documented [48]), allowing them to position themselves at an optimum depth for growth [8]. Similar to *P. alata*, *Tripos* spp. has a high aspect ratio and can distribute their chloroplasts through their horns, or even change their morphology in some species, increasing the surface area to volume ratios and improving light absorption [49]. The large size of *P. alata* and *Tripos* spp.

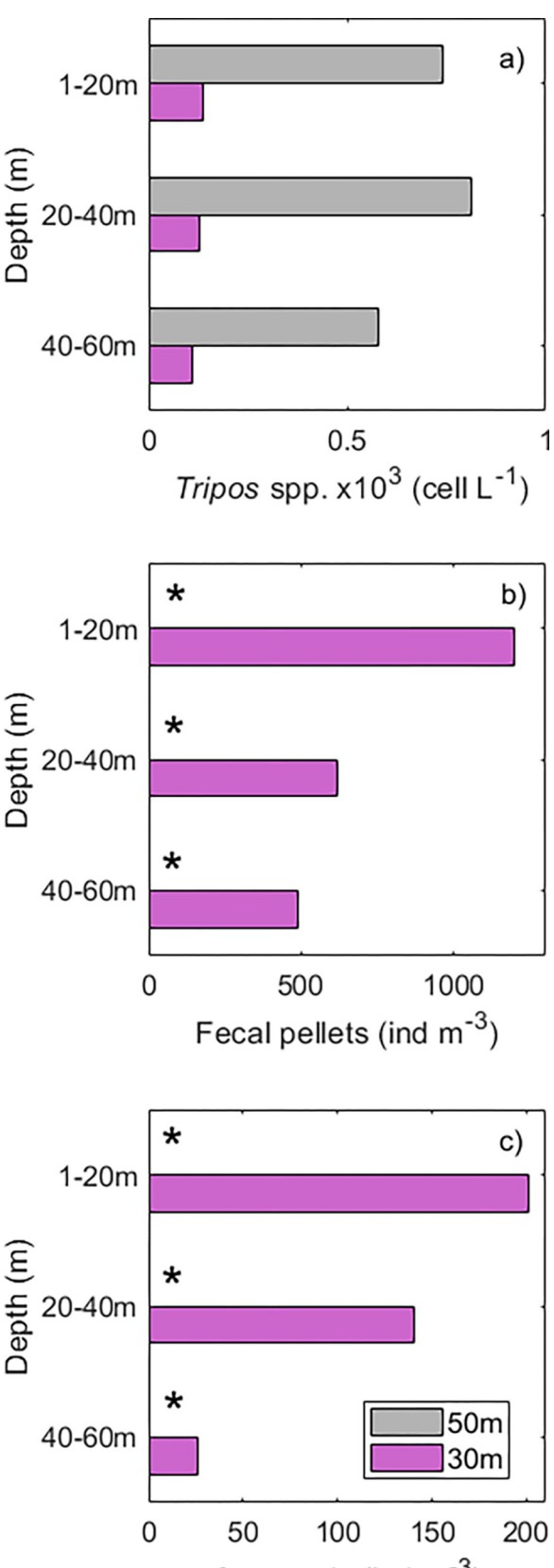

**Fig 7. Silcam data.** a) Abundance of particles identified mostly as a) *Tripos* spp. (×10³ cell L⁻¹), b) fecal pellets and c) copepods (ind m⁻³) derived from the Silcam algorithms and human inspection. The scatterplots with the size spectra of particles imaged per depth interval (1–20 m, 20–40 m and 40–60 m) from the high (red) and low (blue) magnification lenses that most likely represents *Tripos* spp. (100–160 μm), fecal pellets (400–600μm) and copepods (800–1200μm) are shown in S1 and S2 Figs. Asterisks in b) and c) represent lack of data due to technical problems.

may act as a grazing deterrent and be another explanation for their regular occurrence and high abundance in SCMs. Large-sized phytoplankton can physically hinder grazing of small to medium-sized zooplankton [50, 51].

## Zooplankton's role in SCM

Phytoplankton concentrations were high and zooplankton abundances were low in deep mixed waters (50 m). The opposite trend with high zooplankton (both ciliates and copepods) and low phytoplankton abundance was observed in shallow mixed waters (30 m). Simultaneously, fecal pellet and ammonium concentrations were high in regions with high zooplankton abundance. This points at as strong top-down control of phytoplankton standing stocks by ciliates and copepods in shallow mixed waters. The time-series data from the buoy show that waters were stratified for at least one month before the strong wind surge (Fig 3). During the wind event, which coincides with the time of AUV missions and shipboard measurements, inshore waters were, yet, more stratified (MLD ~30 m) and a SCM was observed. The shallow mixing layer depth of inshore waters and the presence of the SCM might have facilitated the encounter rates between predators and their prey, thus providing optimal feeding conditions for ciliate and copepod grazers [52]. Shallow MLD has been hypothesized as a determinant reason for controlling zooplankton grazing rates, since it "confines" the zooplankton-phytoplankton in a limited amount of space and facilitate their encounter [53]. The meso-zooplankton community was dominated by copepods, including *Calanus* spp., which are commonly the most abundant copepod in Norwegian coastal waters during the period from spring to autumn. In contrast to copepods, ciliates can respond rapidly to increases in phytoplankton abundance resulting in instantaneous growth rates and high ciliates standing stocks when phytoplankton availability is high [54]. This strong coupling between primary producers and micrograzers thus explains the high abundance of ciliates off Runde Island. In this study, the high abundance of copepods and ciliates point at a strong top-down control of phytoplankton standing stocks, thus shaping phytoplankton community composition and leading to patchy phytoplankton and zooplankton distribution on sub-mesoscale distance (down to few km). This study reinforces the need for smaller and fine-scale, spatiotemporal sampling to better understand plankton aggregations and patchiness.

## Photo-acclimation in SCM

Despite the low phytoplankton abundance observed in stations with shallow MLD (30 m), [*Chl a*] were higher (both *in-situ* and *in-vitro*) at the subsurface. This suggests that the SCM, in this study, is highly influenced by high [*Chl a*] per cellular ratios, rather than cellular abundance. Variability in cellular *Chl a*:*C* in SCM has largely been attributed to photo-acclimation of phytoplankton to low light and high nutrients near the nutricline, rather than biomass [9]. In this study, high *Chl a*:*C* at the SCM might occur as a response to low light levels (low light-acclimated cells) and high light attenuation ($k_d$) observed in stations with high concentrations of zooplankton and other particles, including fecal pellets.

   Photo-acclimation was also observed in this study through photosynthetic parameters, such as $\sigma_{PII}$' and $\varphi_{PII}$'. The size of $\sigma_{PII}$ in darkness has been suggested to vary according to the

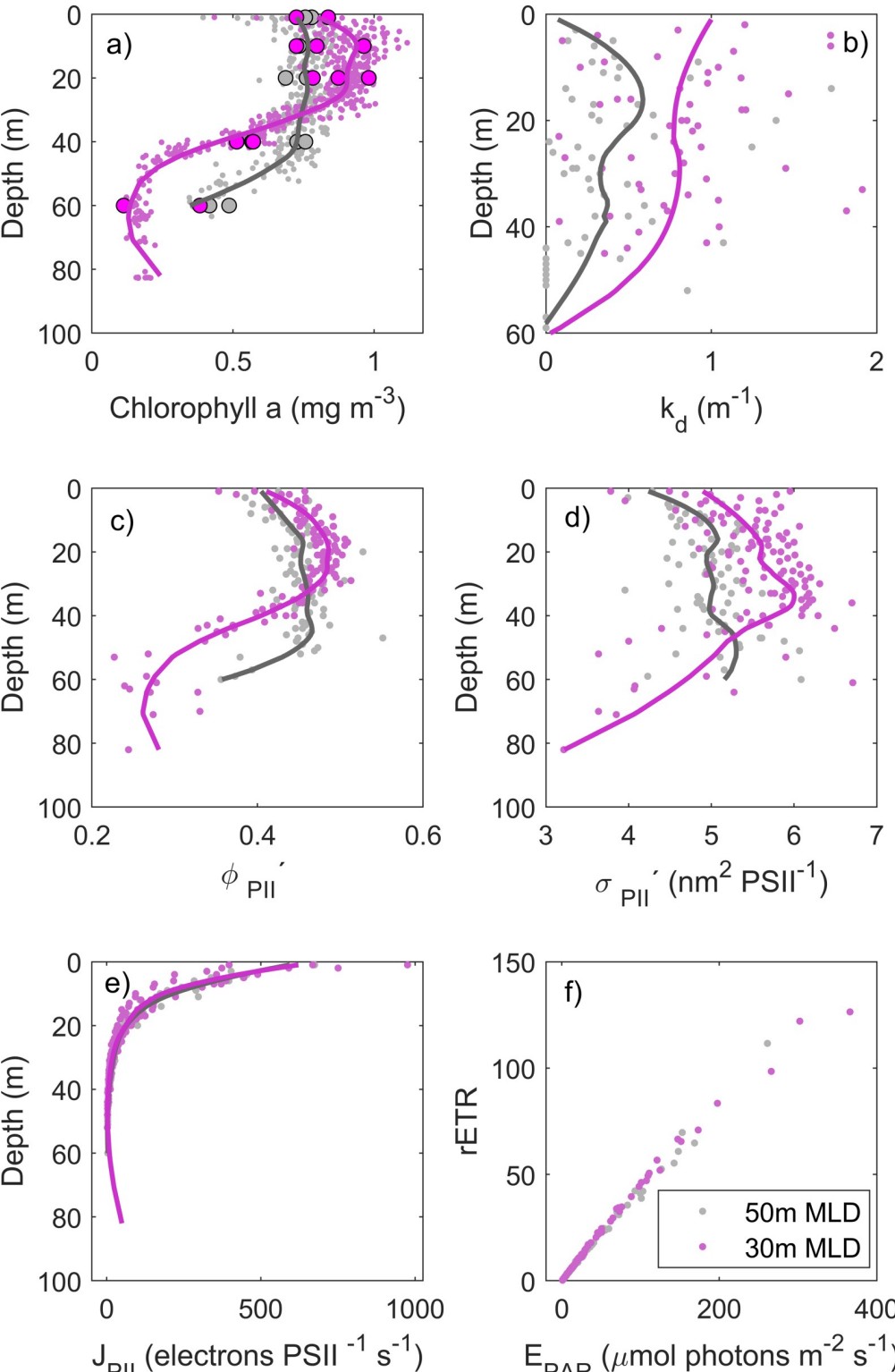

**Fig 8. Photo-physiological data.** Vertical measurements of a) *in-situ* FRRf-derived (*FChla_{FRRf}*, thin dots and *in-vitro* chlorophyll (*Chla_{in-vitro}*, thick dots), b) diffuse light attenuation coefficient (*k_d*), c) Photosystem II efficiency under ambient light (*φ_{PII}*', dimensionless), d) absorption cross-section of PII photochemistry under ambient light (*σ_{PII}*', nm² PSII⁻¹), e) photochemical flux through each open reaction center (*J_{PII}*, electrons PSII⁻¹ s⁻¹) and f) scatterplot of relative electron transport rate (*rETR*) versus irradiance (*E_{PAR}*) derived from the FRRf measurements from pooled stations from areas with deeper mixed layer depth (MLD) (50 m, grey) and shallower MLD (30 m, pink).

taxonomic composition [55] and nutrient status [56]. In this study, taxonomic composition was similar among stations, where the large diatom *P. alata* and several large dinoflagellates of *Tripos* species and other unidentified flagellates were observed. However, their abundance differed within the depth of the mixed layer, where high number of phytoplankton were observed in deep mixed waters (50 m). Change in the size of $\sigma_{PII}$' was, therefore, rather an acclimation response to the wide range of depth irradiances as the cells moved up and down within the mixed layer. To cope with low light availability near the thermocline, phytoplankton cells increased their $\sigma_{PII}$' to absorb more light [57]. Similar results were observed where changes of $\sigma_{PII}$' reflected an acclimation to light intensity and spectral quality varying with depth [56, 57].

$\varphi_{PII}$', which is related to the fraction of absorbed photons that is used in PSII photochemistry, was higher at the subsurface maxima. A high $\varphi_{PII}$' at the SCM suggests optimum conditions, most likely related to the non-saturation of PSII absorbed photons. Thus, part of the absorbed energy was dedicated to photochemistry at the subsurface (Jin et al., 2016), rather than being dissipated as heat through *NPQ*—a phenomenon often observed in the surface. In spite of distinct vertical patterns of $\sigma_{PII}$' and $\varphi_{PII}$', photosynthesis did not reach saturation at all stations and electron flow ($J_{PII}$) and the initial slope from *rETR* versus $E_{PAR}$ curves remained similar among stations. This suggests a high degree of community plasticity and adjusted strategies to changes in the light regime, whereas the overall depth-related flux of electrons in photochemistry was similar between shallow and deep mixed layer regions.

## Conclusions

In this study, a SCM was observed in less exposed stations (inshore), where the mixed layer was shallow (30 m), whilst at more exposed waters (offshore), [*Chl a*] was well distributed throughout the mixed layer (50 m). Strong winds during summer, temporarily disrupted the stratification in offshore waters. Phytoplankton abundances were significantly higher in stations of 50 m MLD (compared to 30 m), where nutrient drawdown occurred throughout the upper 60 m of the water column, suggesting that mixing and nutrient availability boosted phytoplankton growth. Another possible explanation is that disturbance caused by deep mixing, after a prominent period of stratification, might have weakened the typical strong top-down of herbivores by decreasing the encounter rates of phytoplankton-zooplankton. Phytoplankton community composition comprised of *P. alata* and several *Tripos* species, including *T. lineatum and T. fusus*, which are typically observed in stratified, thin SCM layers of temperate shelf seas. Photo-physiological parameters of the phytoplankton community also reflected the vertical hydrographic structure of the water column. A high degree of phytoplankton community plasticity and adjusted strategies to changes in the light regime were observed. The opposite trend in abundance (high zooplankton and low phytoplankton), in addition to high fecal pellet and ammonia concentrations at the upper 20 m suggests a strong grazing pressure of proto- and meso-zooplankton on phytoplankton standing stocks. A shallow MLD might have increased the encounter rates between predators and their prey.

## Supporting information

**S1 Table. FRRf acquisition protocols.** Acquisition protocols (A, B and C) used with different combinations of LED color (blue, green and red) and respective flux ($E_{flux}$, photons nm$^{-2}$ 100 μs$^{-1}$), number of acquisitions collected (n) and average (μ) and standard deviation (±) for the optimum combination of intensity and color of the LED that fully saturates the reaction centers during the first flashlet ($R_{\sigma PII}$).
(PDF)

**S2 Table. *Tripos* species abundances.** Average abundance (cell.L$^{-1}$) of *Tripos* species found at different pooled stations and depth: *T. tripos, T. longipes, T. fusus, T. macroceros, T. lineatum.* * refers to species not found.
(PDF)

**S1 Fig. Silcam data–stations 1 & 2.** Scatterplot of average particle sizes (in equivalent circular diameter, μm) and concentrations (counts/L/μm) derived from the Silcam analyses binned into three depths (1–20 m, 20–40 m and 40–60 m) and for stations with deep mixed layer depth (50 m, stations 1&2). The shaded areas of the scatterplots in the left refer to the size spectra of particles imaged from the high (red) magnification lens that most likely represents *Tripos* spp. (100–160μm). The dashed line in the scatterplot in a) represents the average Junge distributions. Pictures on the right side of the scatterplots in represent collages of particle images from the high (red box) magnification lenses.
(PDF)

**S2 Fig. Silcam data–stations 4 & 5.** Scatterplot of average particle sizes (in equivalent circular diameter, μm) and concentrations (counts/L/μm) derived from the Silcam analyses binned into three depths (1–20 m, 20–40 m and 40–60 m) and for stations with shallow mixed layer depth (30 m, station 4&5). The shaded areas of the scatterplots in the left refer to the size spectra of particles imaged from the high (red) and low (blue) magnification lenses that most likely represents *Tripos* spp. (100–160μm), fecal pellet (400–600μm) and copepods (800–1200μm). The dashed line in the scatterplot in a) represents the average Junge distributions. Pictures on the right side of the scatterplots represent collages of particle images from the high (red box) and low (blue box) magnification lenses.
(PDF)

**S1 File. Raw data from sensor and plankton counts.**
(XLSX)

## Acknowledgments

We thank the crew of the *RV Gunnerus* and the technicians and scientists involved in the cruise for their support. Many thanks to the staff of Runde Environmental Center for their logistical help with sampling and accommodation during field work. Buoy data were collected by Fugro Norway AS for the Norwegian Public Roads Administration and published by the Norwegian Meteorological Institute.

## Author Contributions

**Conceptualization:** Glaucia M. Fragoso.

**Formal analysis:** Glaucia M. Fragoso, Emlyn J. Davies, Trygve O. Fossum, Jenny E. Ullgren, Sanna Majaneva, Nicole Aberle.

**Funding acquisition:** Martin Ludvigsen, Geir Johnsen.

**Investigation:** Glaucia M. Fragoso.

**Methodology:** Glaucia M. Fragoso, Emlyn J. Davies, Trygve O. Fossum, Jenny E. Ullgren, Sanna Majaneva, Nicole Aberle, Martin Ludvigsen.

**Supervision:** Martin Ludvigsen, Geir Johnsen.

**Validation:** Glaucia M. Fragoso, Trygve O. Fossum, Sanna Majaneva, Nicole Aberle.

**Visualization:** Glaucia M. Fragoso, Emlyn J. Davies, Trygve O. Fossum.

**Writing – original draft:** Glaucia M. Fragoso, Emlyn J. Davies, Jenny E. Ullgren, Sanna Majaneva, Nicole Aberle.

**Writing – review & editing:** Glaucia M. Fragoso, Emlyn J. Davies, Trygve O. Fossum, Jenny E. Ullgren, Sanna Majaneva, Nicole Aberle, Geir Johnsen.

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
