## [Decision Letter · Decision Letter 0]

9 Jun 2022

PONE-D-22-12849Dissecting plankton patchiness through autonomous platforms and in-situ optical sensors.PLOS ONE

Dear Dr. Fragoso,

Thank you for submitting your manuscript to PLOS ONE. After careful consideration, we feel that it has merit but does not fully meet PLOS ONE’s publication criteria as it currently stands. Therefore, we invite you to submit a revised version of the manuscript that addresses the points raised during the review process.

We look forward to receiving your revised manuscript.

Kind regards,

Emmanuel S. Boss

Academic Editor

PLOS ONE

Journal Requirements:

3. We noted in your submission details that a portion of your manuscript may have been presented or published elsewhere. 

"Partial data, including one of the AUV missions and tracking of subsurface chlorophyll maxima has been published in Fossum et al 2019 Science Robotics. Fossum et al is a technical paper focused on the adaptive sampling of subsurface chlorophyll maxima. Our article use parts of his data, such as the 3D plots of chlorophyll distributions (Fig 3b,d) and parts of the Silcam image (Fig 6, 30m MLD only - although we pooled the data at different depths, so it is not an identical figure) in region B. In our manuscripts, we add unpublished data (data from region A) to compare the contrasting patterns of patchiness between the two regions. Besides, we add unpublished photo-physiological data from the FRRf (Fig 7), nutrient data (Fig 4) and phytoplankton and meso-zooplankton counts (Fig 5). While Fossum et al is a technical and methodological paper for engineers, our paper is more interdisciplinary and the discussion is focused on the bio-physical interactions that shape plankton patchiness."

Additional Editor Comments:

Dear authors,

As you see in the reviewer comments, the two reviewers have come to opposite conclusions regarding your paper. Reviewer 1 think it is nearly ready to be published while reviewer 2 has identified important major issues with your paper (including formatting issues associated with your figures and interpretation of the data). Please read carefully all comments as the reviewers want to help you get the best work possible to the public.

Reviewers' comments:

Reviewer's Responses to Questions

**Comments to the Author**

1. Is the manuscript technically sound, and do the data support the conclusions?

Reviewer #1: Yes

Reviewer #2: No

2. Has the statistical analysis been performed appropriately and rigorously? 

Reviewer #1: N/A

Reviewer #2: No

3. Have the authors made all data underlying the findings in their manuscript fully available?

Reviewer #1: Yes

Reviewer #2: No

4. Is the manuscript presented in an intelligible fashion and written in standard English?

Reviewer #1: Yes

Reviewer #2: Yes

5. Review Comments to the Author

Reviewer #1: This manuscript reports the results of detailed surveys of the Norwegian coastal sea. It is an exemplary combination of conventional oceanographic sampling, AUV profiling and “in situ” methods. The results from the two contrasting study areas highlight the importance of episodic mixing events in the productivity of shelf seas and outline key adaptations, including photo-physiology of the dominant species involved. While the study highlights the importance of subsurface chlorophyll maxima (SCM), it also draws attention to the significance of episodic mixing events in recharging nutrients. As such the study is a really useful addition to the literature on the biogeochemistry of coastal seas.

On the whole, the manuscript is put together well, however there is some unnecessary confusion regarding the labelling of the different study areas and the methods of “pooling” of data. More detailed recommendations regarding this are given below, together with a list of minor points of phrasing or ambiguity that require correction.

“Pooled stations” and location of Fig 3 data – these issues are confusing.

This requires some revision of Figures 1 and 3 and captions and in the text.

To avoid confusion, the survey areas should be designated A and B at the outset and should be indicated as such on Figure 1.

The “pooling” concept is currently mentioned in the manuscript in the following order:

In the “Materials and Methods” section under the “Sampling” heading, it is stated that vertical profiles were taken at five stations, presumably the stations numbered 1 through 5 on Figure 1.

“Pooling” is first mentioned in the caption to Figure 1:

“c, Detailed map showing the area where the AUV mission was conducted and stations pooled based on similar characteristics (for more details, see results section)”

In the key to Fig. 1 “pooled stations” has black line, whereas the stations appear to be enclosed with lines of different colour (grey and red). In the absence of further clarification, this is confusing. There is a small black line to the SE of station 5, but I assume this is not what is meant. It would be better to indicate “pooled stations” in the key with a closed loop.

In fact, the “more details” are given, not in the results section (as stated in the caption to Fig. 1), but in the “Materials and Methods” section under the “Data Analyses” heading, where it is stated that the station profiles:

“were pooled (averaged by depth) as stations 1 and 2, and stations 3, 4 and 5. Pooling of these stations were (sic) based on similar hydrographical structure ….”

It would be clearer if the “pooling” or averaging was clarified earlier.

The case for “pooling” would be stronger if the salinity/ temperature/ chlorophyll – depth profiles of the five individual stations were shown in the supporting information.

This confusion is further compounded by the lack of adequate detail in Figure 3 and its caption. Figure 3a and 3b are the AUV profiles and are labelled A and B in Fig. 3, but are not labelled as such on Fig. 1. In the caption to Fig. 3, the two AUV survey areas are designated “A-left and B-right” but there is no A or B label on Fig 1. Do A and B refer to the NE and SW AUV surveys respectively? This needs to be clarified in the figures and captions.

Do Fig 3c and 3d represent the multiple AUV profiles from the two AUV profile areas? It would help to clarify this – also the caption incorrectly has these as Fig 3 b,c.

Finally, the single panel in Fig 3e contains what I assume to be the shipboard profiles

from both pooled stations 1 and 2 and from pooled stations 3, 4 and 5. It would be clearer have these as two separate panels beneath the corresponding AUV survey panels above, depicting results from areas A and B.

Also, it looks as if the plots in Fig 3e contain multiple superimposed profiles and not and averaged profile for the two pooled areas (as suggested in the text). This should be clarified.

Minor points:

Line 19: replace “at” with “off”

Line 24: should this not be “observed above the base of the mixed layer”?

Line 27: does “twice lower” mean “half”? – if so best to change

Line 37: insert “a” after suggesting

Line 38: should read “Our results emphasize”

Line 46: delete “by”

Line 54: delete “water layer, namely”

Line 60: add “the development of” before SCMs

Line 62 replace “is” with “are”

Line 71: replace “vastly” with “extensively”

Line 80: insert “the dinoflagellate” before Tripos – to be consistent with diatom mention

Line 82: insert “the” before “main”

Line 122: “close in time” – vague

Line 153: “mode strategies” – meaning unclear

Line 157: unclear to what this velocity refers

Line 162: double bracket

Line 165: replace “information” with “determination”

Line 226 delete “This means that”

Lines 238-239; need to mention areas A and B – se detailed comments above.

Line 330: “(and vice versa)” – unclear to what is meant here

Line 337: insert “for” before “which”

Line 409: insert “an” before “episodic”

Line 410: insert “an” before “event”

Line 413: should be “layers”

Line 414: should be “dinoflagellates”

Line 422: replace “on” with “within”

Line 423: insert “a” after “Such”

Line 437: replace “has” with “have”

Line 443: should be “suggests”

Lines 452-454: confused sentence – need re-write

Line 456: replace “mixing” with “mixed”

Line 460: should be “conditions”

Line 479: should be “ciliate”

Line 487: delete “even”

Line 497: replace “was noticed” with “occurred”

Line 499: delete “were”

Line 506: should be “A shallow…”

Reviewer #2: Review of the manuscript entitled « Dissecting plankton patchiness through autonomous platforms and in-situ optical sensors” presented for consideration in Plos One by Fragoso et al.

General remarks.

In this manuscript, Fragoso and co-authors reports the use of various instruments during a sampling effort in northern Norway, following a sampling from inshore to offshore waters. During this sampling, AUV have been deployed to measure classical T,S, chla measurements while other instruments were deployed from the boat itself, including an imaging instrument (silcam) and an optical one (FRRf). The course of the sampling was perturbated by a strong mixing event (without clear definition of its timing compared to the cruise. Was it before or during? If during, I would say between station 2 and 3 but this remains a guess), which adds to the duality offshore/inshore a second level of duality (mixed vs non-mixed) which do not ease the interpretation.

The aim of the dataset gathered and the purpose of the sampling cruise are not clearly defined and may look as the result of a test deployment (or maybe it was not the case and this was omitted or got aborded as a result of the “strong mixing” happening between the two group of stations). Consequently, the observations do not clearly target any precise process and are hardly demonstrating clear interactions between the different measurements conducted (see below on the mismatch between some measurements cross-interpreted). Moreover, there is a clear decoupling between the title (patchiness is nearly not studied nor dissected here, only the presence/absence of a sub-surface chlorophyl maxima -SCM -is reported), the introduction (with a clear demonstration of the importance of different processes to control the SCM) and the latter lack of discussion or exploitation of those processes in regards to the results. Even the AUV results (done to “track the tri dimensional chl a patchiness”, line 75) are only used to confirm the mixed vs non mixed nature of both environments (fig 3) but is never used to explore the said patchiness.

This said, at sea everything happens and the best should be tried to use the collected data. However, the manuscript also suffers from three large weaknesses:

1) First of all, the quality of all figures is just unacceptable as it. They arrived really pixelated and at least Fig 6 is purely not usable: plankton images are not visible: is it really plankton that we are supposed to see? Fonts are too small to not be blurred pixels- on a full-page figure-. Some axes do not make sense (see vertical axe on figure 5e) or are missing critical information (where are A and B zones on figure 1; how are located currents mentioned in the text?). All this also cast doubts also on the quality of image collected and how they are identified or used, and is not helped by the fact that it seems that the images were only classified by an algorithm (line 167-171) without human expertise and without estimation of the efficiency of classification. This is clearly not acceptable without an estimation of the error made or at least a verification of a subset of the image collected.

2) Most of the results and conclusions relies on the discrepancies between counts of large phytoplankton species (Tripos from both Silcam and microscope counts; Proboscia and ciliates from microscopy) and other measurements relying on in-situ Chla observed by optical means of through filtration on GF/F filters (0.7µm) and chla measurements. However, this left most of the phytoplankton, and their associated chlorophyll a or optical photosynthetic parameters, from 0.7µm to several tens of microns unmeasured by counts. Therefore without at least realizing this discrepancy and recognizing this bias, any conclusion based on such uncoupling between counts and chlorophyll / optical measurements are purely speculative if not purely wrong…. since they could not be coupled together. At least using flowcytometry to fill this size gap should have been conducted, but other methods also exist.

3) Finally, most of the discussion seems to be either very speculative or to ignore potential other hypothesis. Indeed, most of the results are interpreted as if the two zones were relatively similar in their structure prior to the wind mixing. Therefore, all explanations to their potential differences are interpreted to be the result of the mixing event (that have happened during or before the cruise? This is not specified…) and to their protected/exposed nature. However, without measurements showing that they were similar in their nature before such interpretation is just purely speculative. There is an alternative explanation: that they were already different prior to the mixing. Indeed, most of the observed uncoupled associations between nutrients/phytoplankton/zooplankton seems to be coherent with a top-down control (from zooplankton on large phytoplankton? From larger predators on zooplankton?). Without exploring other hypothesis and processes which may have caused the observed differences between the two sites, the discussion is not only speculative but also very partial and oriented in the presented conclusions.

Because of the above-mentioned weaknesses, I could not recommend publication, at least with a clearly major change in results uses and discussion.

Detailed remarks

Title: I am not sure after having read the full manuscript that any results were really exploiting the “patchiness” aspects of the data (except maybe the fact that there is a deep chlorophyl maxima), not to talk about “dissecting” this patchiness. On the same aspect, I was really enthusiast about the title which let me think that in-situ optical sensors were used onboard autonomous platforms (gliders)…. While in fact it is not at all the case. The abstract did not clearly correct this and this is only in the material and methods that I finally understood that it was not the case. Because of those two points, the title is misleading and should be changed or corrected.

Line 19: adaptative sampling should be understandable by someone that are not familiar with gliders deployments: please explain here.

Line 26: “exposed” exposed to what? (maybe just used inshore/offshore)

27: if sub-surface chlorophyll maxima is correlated with a decrease of abundance of those two large phytoplankton cells, maybe this should indicate that other phytoplankton are responsible for this:

Line 91 (section study area): most of this section present why the study area is a great study site in term of scientific background (reduction of seabird etc…). Therefore I suggest moving most of this section to introduction.

Line 95-96: Could you please indicate those two currents on the Fig 1 to guide the reader?

Line 118: Fig 1 should be called since there.

Figure 1: A and B boxes (used latter) should be indicated here. The line style and line width of those boxes should be darker and larger to be easily spotted (this is not the case here). Consider as well that the color choice for depth is potentially bad: impossible to disentangle depth 225 from depth 600. Additionally, I guess those depth should be rather explained with range of depth (e.g. 200-300; 300-900m), except if the Norwegian seafloor is peculiar in its bathymetry.

In what extend data collected with the boat and with the glider could be compared (notably true since the glider in box A operated in deeper waters (225? 600?) than the close-by sampling stations (125m).

Line 143: AUV date deployment? Was it the same AUV deployed two times over short period or two deployed over a longer period?

Line 167-171: automatic recognition is often not enough (but see https://doi.org/10.1146/annurev-marine-041921-013023 ). Were the images manually inspected ? Was the efficiency of the image recognition evaluated by any means? The images provided as examples in figure 6 do not allow any understanding on the quality of the process. How much images were acquired? How much classified/ verified by an human expert?

Line 180: ST is an acquisition protocol not a measurement (line 176), please rephrase.

Line 186: nice explanation, but why not having repeated the same kind of explanations for other parameters bellow. (And no, table 1 does not further demystify those)

Line 210: this protocol leads to measure chl a in >0.7µm organisms. Which could potentially explain much difference in between some given parameters (see bad coupling with large phytoplankton reported in abstract). Having this compared with Lugol staining and microscopic counts without explaining what count was operated do not help the reader. What minimal size was considered? Or did only the two target phytoplankton species were counted (which have size >20µm at least)?

Line 216-219: Ethanol fixation can cause shrinkage of organisms and potentially bias the samples. How this may bias the results or did only a count was done?

Line 243: why did you pooled those samples? Was the station 5 not acquired? If yes, this should be mentioned since the material and methods. From the fig 1, it seems more logical to pool station 4&5 to compare with the glider, station 3 been right in between the two glider areas. I do understand that this grouping was made on hydrographic structure … which is never shown on a per profile basis.

Results:

Line 266-267:there is no way to say that the vertical structure of AUV and CTD show similar patterns here, and it is impossible to judge this without plotting those side by side with similar measurements: here temperature, chlorophyl and salinity are shown from AUV while density is displayed only from boat measurements (no temperature/salinity, chl a?).

Is the difference of MLD due to the inshore/offshore nature of the stations or did the wind mixing occurred somehow between station 2 and 3?

Line 290: add space

Line 302 and onward (not a comment but important for latter comments): here is related that deeper-mixed waters have less nitrates/ higher (large) phytoplankton (but no SCM) and lower zooplankton. On the contrary shallow mixed waters have higher nutrients, low (large) phytoplankton (but high Chla) and high zooplankton. All this looks like the usual symptoms of an ecosystem under top-down control.

Discussion

Line 410-411: very speculative, from the results obtained, this could also be due to a trophic cascade (top down effect) rather than the bottom up effect proposed. But without temporal observations, this is hard to say. Did the AUV were at sea a long time and could provide such temporal vision?

Line 431: reference 41 format in different.

Line 442-458: this part is really speculative since those observations could be the results of several inhomogeneous measurements: Chla both seen by sensors and measured on filters is usually from small size fraction (filtered onto 0.2µm filter). The fact to have measured low abundance of two large phytoplankton is not an indication that all phytoplankton were low (and higher Kd seem to indicate larger particles load….. usually correlated with small phytoplankton or sediments). All this part relies on the fact that all phytoplankton are supposed to be represented by the two target species identifies and counted on microscope. Since this is very likely not the case, the whole part needs to be modified to take this into account.

Another important thing is that those high chla/ low phytoplankton have large amounts of zooplankton (which prefers larger preys) and may help in amplifying this discrepancy.

Line 472-484: all this explanation (on differences between zooplankton and phytoplankton) relies on the assumption that all zooplankton reacts like ciliates (with rapid respond to phytoplankton bloom). However, half of zooplankton measured are instead copepods, which does have a lifetime of the level of months and then could not respond within the (ungiven) timeframe between mixing and sampling. Therefore, this part sound really speculative. Moreover, it relies on the untold assumption that conditions were about similar before the mixing, which is again unlikely to be the case. One other explanation is that deeply mixed waters stations are naturally low in abundance of zooplankton…. And this could be due to higher predation onto zooplankton (fishes?). Since the described process sounds really in line with top-down control of those ecosystems, I suggest the authors to rethink their results in this respect and not necessarily try to explain those by bottom up control processes (note that both could happens together).

Figure 5: the bottom panel (e) have a depth axis together with taxonomy

Figure 6: this figure has really a bad quality and do not allow to neither read correctly the axis not see the images that are supposed to represent plankton species … is the present state, this is not possible at all and could not be used at all to understand results. Such low quality figure (not that all the other ones were badly pixeled too) is unacceptable and could not be reviewed.

6. PLOS authors have the option to publish the peer review history of their article (what does this mean?). If published, this will include your full peer review and any attached files.

Reviewer #1: No

Reviewer #2: No

---

## [Author Response · Author response to Decision Letter 0]

12 Jul 2022

We thank the reviewers for their comments and suggestions, which we feel that have greatly improved the manuscript. Below we respond to each comment in detail. RC refers to “Reviewer’s Comments” and AC to “Author’s comments”. Changes are in track changes.

RC1: Reviewer #1: This manuscript reports the results of detailed surveys of the Norwegian coastal sea. It is an exemplary combination of conventional oceanographic sampling, AUV profiling and “in situ” methods. The results from the two contrasting study areas highlight the importance of episodic mixing events in the productivity of shelf seas and outline key adaptations, including photo-physiology of the dominant species involved. While the study highlights the importance of subsurface chlorophyll maxima (SCM), it also draws attention to the significance of episodic mixing events in recharging nutrients. As such the study is a really useful addition to the literature on the biogeochemistry of coastal seas.

On the whole, the manuscript is put together well, however there is some unnecessary confusion regarding the labelling of the different study areas and the methods of “pooling” of data. More detailed recommendations regarding this are given below, together with a list of minor points of phrasing or ambiguity that require correction.

“Pooled stations” and location of Fig 3 data – these issues are confusing.

This requires some revision of Figures 1 and 3 and captions and in the text.

To avoid confusion, the survey areas should be designated A and B at the outset and should be indicated as such on Figure 1.

The “pooling” concept is currently mentioned in the manuscript in the following order:

In the “Materials and Methods” section under the “Sampling” heading, it is stated that vertical profiles were taken at five stations, presumably the stations numbered 1 through 5 on Figure 1.

“Pooling” is first mentioned in the caption to Figure 1:

“c, Detailed map showing the area where the AUV mission was conducted and stations pooled based on similar characteristics (for more details, see results section)”

In the key to Fig. 1 “pooled stations” has black line, whereas the stations appear to be enclosed with lines of different colour (grey and red). In the absence of further clarification, this is confusing. There is a small black line to the SE of station 5, but I assume this is not what is meant. It would be better to indicate “pooled stations” in the key with a closed loop.

AC1: For clarification, we removed the pooling from Figure 1 and added the labels A and B referring to the AUV sampling box. 

RC2: In fact, the “more details” are given, not in the results section (as stated in the caption to Fig. 1), but in the “Materials and Methods” section under the “Data Analyses” heading, where it is stated that the station profiles:

“were pooled (averaged by depth) as stations 1 and 2, and stations 3, 4 and 5. Pooling of these stations were (sic) based on similar hydrographical structure ….”

It would be clearer if the “pooling” or averaging was clarified earlier.

The case for “pooling” would be stronger if the salinity/ temperature/ chlorophyll – depth profiles of the five individual stations were shown in the supporting information.

AC2: We removed the pooling from figure 1 and added the vertical profiles of each station at Figure 4.

RC3: This confusion is further compounded by the lack of adequate detail in Figure 3 and its caption. Figure 3a and 3b are the AUV profiles and are labelled A and B in Fig. 3, but are not labelled as such on Fig. 1. In the caption to Fig. 3, the two AUV survey areas are designated “A-left and B-right” but there is no A or B label on Fig 1. Do A and B refer to the NE and SW AUV surveys respectively? This needs to be clarified in the figures and captions.

AC3: We have now added the label A and B in Figure 1.

RC4: Do Fig 3c and 3d represent the multiple AUV profiles from the two AUV profile areas? It would help to clarify this – also the caption incorrectly has these as Fig 3 b,c.

AC4: We have included information in the caption that the Fig 4 represents multiple profiles from the AUV and fixed the figure labels in the caption.

RC5: Finally, the single panel in Fig 3e contains what I assume to be the shipboard profiles

from both pooled stations 1 and 2 and from pooled stations 3, 4 and 5. It would be clearer have these as two separate panels beneath the corresponding AUV survey panels above, depicting results from areas A and B.

AC5: We have now fixed the caption and added the profiles for each station as suggested by the reviewer.

RC6: Also, it looks as if the plots in Fig 3e contain multiple superimposed profiles and not and averaged profile for the two pooled areas (as suggested in the text). This should be clarified.

AC6: We have now plotted each profile separately (see Fig. 4e,f).

Minor points:

RC7: Line 19: replace “at” with “off”

AC7: Fixed (Line 22, track changes version).

RC8: Line 24: should this not be “observed above the base of the mixed layer”?

AC8: Yes. This is now fixed (Line 29, track changes version).

RC9: Line 27: does “twice lower” mean “half”? – if so best to change

AC9: Yes. This is now fixed (Line 33, track changes version).

RC10: Line 37: insert “a” after suggesting

AC10: Fixed (Line 44, track changes version).

RC11: Line 38: should read “Our results emphasize”

AC11: Fixed (Line 45, track changes version).

RC12: Line 46: delete “by”

AC12: Fixed (Line 52, track changes version).

RC13: Line 54: delete “water layer, namely”

AC13: Fixed (Line 60, track changes version).

RC14: Line 60: add “the development of” before SCMs

AC14: Fixed (Line 66, track changes version).

RC15: Line 62 replace “is” with “are”

AC15: Fixed (Line 69, track changes version).

RC16: Line 71: replace “vastly” with “extensively”

AC16: Fixed (Line 77, track changes version).

RC17: Line 80: insert “the dinoflagellate” before Tripos – to be consistent with diatom mention

AC17: It is now included (Line 93, track changes version).

RC18: Line 82: insert “the” before “main”

AC18: It is now inserted (Line 95, track changes version).

RC19: Line 122: “close in time” – vague

AC19: changed to “at approximate time” (Line 142, track changes version).

RC20: Line 153: “mode strategies” – meaning unclear

AC20: we removed “mode strategies” (Line 190, track changes version).

RC21: Line 157: unclear to what this velocity refers

AC21: We have now changed this sentence for clarification (Line 193, track changes version).

RC 22: Line 162: double bracket

AC22: Fixed (Line 200, track changes version).

RC23: Line 165: replace “information” with “determination”

AC23: Changed (Line 203, track changes version).

RC24: Line 226 delete “This means that”

AC24: Deleted (Line 294, track changes version).

RC25: Lines 238-239; need to mention areas A and B – se detailed comments above.

AC25: The areas A and B are now mentioned (Line 310, track changes version). 

RC26: Line 330: “(and vice versa)” – unclear to what is meant here

AC26: This was removed (Line 416, track changes version).

RC27: Line 337: insert “for” before “which”

AC27: Inserted (Line 424, track changes version).

RC 28: Line 409: insert “an” before “episodic”

AC 28: Inserted (Line 498, track changes version).

RC29: Line 410: insert “an” before “event”

AC29: Inserted (Line 499, track changes version).

RC30: Line 413: should be “layers”

AC30: Changed (Line 503, track changes version).

RC31: Line 414: should be “dinoflagellates”

AC31: Changed (Line 504, track changes version).

RC 32: Line 422: replace “on” with “within”

AC32: Changed (Line 512, track changes version).

RC33: Line 423: insert “a” after “Such”

AC33: Changed (Line 514, track changes version).

RC34 Line 437: replace “has” with “have”

AC34: Changed (Line 527, track changes version).

RC35: Line 443: should be “suggests”

AC35: Changed (Line 563, track changes version).

RC36: Lines 452-454: confused sentence – need re-write

AC36: Sentence was re-written for clarification (Lines 572-577, track changes version).

RC37: Line 456: replace “mixing” with “mixed”

AC37: Fixed (Line 579, track changes version).

RC38: Line 460: should be “conditions”

AC38: Corrected (Line 584, track changes version).

RC39: Line 479: should be “ciliate”

AC39: Fixed.

RC40: Line 487: delete “even”

AC40: Deleted (Line 555, track changes version).

RC41: Line 497: replace “was noticed” with “occurred”

AC41: Changed (Line 621, track changes version).

RC42: Line 499: delete “were”

AC42: Deleted (Line 623, track changes version).

RC43: Line 506: should be “A shallow…”

AC43: Changed (Line 630, track changes version).

Reviewer #2: Review of the manuscript entitled « Dissecting plankton patchiness through autonomous platforms and in-situ optical sensors” presented for consideration in Plos One by Fragoso et al.

RC 44: General remarks.

In this manuscript, Fragoso and co-authors reports the use of various instruments during a sampling effort in northern Norway, following a sampling from inshore to offshore waters. During this sampling, AUV have been deployed to measure classical T,S, chla measurements while other instruments were deployed from the boat itself, including an imaging instrument (silcam) and an optical one (FRRf). The course of the sampling was perturbated by a strong mixing event (without clear definition of its timing compared to the cruise. Was it before or during? If during, I would say between station 2 and 3 but this remains a guess), which adds to the duality offshore/inshore a second level of duality (mixed vs non-mixed) which do not ease the interpretation.

AC44: We understand the reviewer’s confusion. For clarification, we now included Table 1 (which was previously in the Supplementary material) in the main text. This table resumes the time and days when sampling occurred. We also emphasized in Fig 3 (see now fig 3d) and in the results (lines 340 – 347, track changes version) that the storm occurred throughout the sampling days. We also added in these lines that the difference in mixed layer depth could possibly be caused by the fact that offshore areas are more exposed to wind than inshore areas, which are surrounded by islands. 

RC45: The aim of the dataset gathered and the purpose of the sampling cruise are not clearly defined and may look as the result of a test deployment (or maybe it was not the case and this was omitted or got aborded as a result of the “strong mixing” happening between the two group of stations). Consequently, the observations do not clearly target any precise process and are hardly demonstrating clear interactions between the different measurements conducted (see below on the mismatch between some measurements cross-interpreted).

AC45: For clarification, we have now addressed the aim of our research in the introduction (line 78-103, track changes version). The target processes are now pointed out in this section of introduction as well as considered in the discussion: potential bottom-up (lines 493-510, track changes version) and top-down processes (531-557, track changes version) and photo-acclimation (lines 560-592, track changes version) 

AC46: Moreover, there is a clear decoupling between the title (patchiness is nearly not studied nor dissected here, only the presence/absence of a sub-surface chlorophyl maxima -SCM -is reported), the introduction (with a clear demonstration of the importance of different processes to control the SCM) and the latter lack of discussion or exploitation of those processes in regards to the results. Even the AUV results (done to “track the tri dimensional chl a patchiness”, line 75) are only used to confirm the mixed vs non mixed nature of both environments (fig 3) but is never used to explore the said patchiness.

AC46: The reviewer made a good point. To clarify the information of the paper, we changed the tittle to: “Contrasting phytoplankton-zooplankton distributions observed through autonomous platforms, in-situ optical sensors and discrete sampling” 

RC47: This said, at sea everything happens and the best should be tried to use the collected data. However, the manuscript also suffers from three large weaknesses:

1) First of all, the quality of all figures is just unacceptable as it. They arrived really pixelated and at least Fig 6 is purely not usable: plankton images are not visible: is it really plankton that we are supposed to see? Fonts are too small to not be blurred pixels- on a full-page figure-. Some axes do not make sense (see vertical axe on figure 5e) or are missing critical information (where are A and B zones on figure 1; how are located currents mentioned in the text?). 

AC47: We have now improved the quality of all figures and increased the font size. We fixed the vertical axis on figure 6. We have now added Figure 2 as an example of collage images of particles from the high and low magnification lenses. We included what previously was Figure 6 now in the supplementary material, where it shows better resolution and covers the whole page. For Figure 1, we added the labels A and B for the AUV survey box. We have now added the currents at Fig. 1a. 

The figures are in high quality, but they lose their quality when uploaded separately in the Plos One Editorial Manager. You can check them in high quality if you download each figure from the generated pdf. We have also added a word document where we put our figures and they look in much better quality in the printed pdf.

RC 48: All this also cast doubts also on the quality of image collected and how they are identified or used, and is not helped by the fact that it seems that the images were only classified by an algorithm (line 167-171) without human expertise and without estimation of the efficiency of classification. This is clearly not acceptable without an estimation of the error made or at least a verification of a subset of the image collected.

AC48: Thank you for pointing out the issue with the images. The image quality in figure 6 was poor due to rendering, rather than poor image quality from the SilCam. This has now been rectified in the revised version (see Fig 2). We have added more detailed regarding the method and verification steps used. We added relevant citations that support the use of this method (lines 205-217, track changes version).

Note that microscopic counts from Lugol samples (for Tripos) and plankton nets (for main zooplankton groups) were also added in this manuscript to support the information provided by the Silcam regarding contrast phytoplankton-zooplankton abundances found in both areas (inshore vs. offshore). 

RC49: 2) Most of the results and conclusions relies on the discrepancies between counts of large phytoplankton species (Tripos from both Silcam and microscope counts; Proboscia and ciliates from microscopy) and other measurements relying on in-situ Chla observed by optical means of through filtration on GF/F filters (0.7µm) and chla measurements. However, this left most of the phytoplankton, and their associated chlorophyll a or optical photosynthetic parameters, from 0.7µm to several tens of microns unmeasured by counts. Therefore without at least realizing this discrepancy and recognizing this bias, any conclusion based on such uncoupling between counts and chlorophyll / optical measurements are purely speculative if not purely wrong…. since they could not be coupled together. At least using flowcytometry to fill this size gap should have been conducted, but other methods also exist.

AC49. We have now clarified in the text (see lines 275-283, track changes) that microscopic counts of phytoplankton > 4 µm from lugol samples was also included. The abundance of flagellates (cryptophytes, dinoflagellates other than Tripos spp. and other unidentified flagellates) is included in Fig 6d and clarified in the figure legend (see lines 404-407, track changes). Thus, the uncoupling between chlorophyll and phytoplankton abundances is neither speculative nor wrong. We discuss the influence of photo-acclimation in explaining this uncoupled observation (line 562-569). 

RC50: 3) Finally, most of the discussion seems to be either very speculative or to ignore potential other hypothesis. Indeed, most of the results are interpreted as if the two zones were relatively similar in their structure prior to the wind mixing. 

AC50: We have now included the top-down hypothesis following the suggestions of the reviewer (see comments below). The high copepod and ciliate abundances point at a strong top-down control of phytoplankton in the upper water column at the inshore stations. This assumption is further supported by the high ammonium and fecal pellets concentrations at the inshore stations compared to the more offshore stations. We have revised the discussion section now to make it clear that we do not interpret the two zones being very similar prior to the storm. 

We are sorry if our formulations sounded misleading. What we state is that the two zones are different in terms of hydrographical characteristics and biotic activities. In this study, we compare the contrasting zones referring to the different hydrographic characteristics. We state that offshore waters were subject to wind stress leading to mixing of the water column (based on AUV, buoy and shipboard profiles data). Based on the buoy data, the offshore waters were characterised by a stratified water column for at least one month (see Fig 3) before mixing due to high wind stress. We state that the mixing of the water column most likely resulted in an introduction of nutrients to the upper water column, thus stimulating phytoplankton growth in the areas of deep mixing (bottom-up control). As stated above, this contrasts with inshore waters, where high zooplankton (copepod and ciliate) abundances, high ammonium and fecal pellets concentrations and stratification occurred while phytoplankton concentration were comparably low (top-down control). 

RC51: Therefore, all explanations to their potential differences are interpreted to be the result of the mixing event (that have happened during or before the cruise? This is not specified…) and to their protected/exposed nature. However, without measurements showing that they were similar in their nature before such interpretation is just purely speculative. There is an alternative explanation: that they were already different prior to the mixing.

AC51: We have now specified the period of sampling in relation to the storm (AC44). We clarified now that we did not interpret the data as if the two zones were similar prior to the storm (see AC50). We try to link the hydrographic data with potential top-down and bottom-up processes based on the available data to explain the contrasting observations between the two zones (see AC50, lines 484-549).

RC52: Indeed, most of the observed uncoupled associations between nutrients/phytoplankton/zooplankton seems to be coherent with a top-down control (from zooplankton on large phytoplankton? From larger predators on zooplankton?). Without exploring other hypothesis and processes which may have caused the observed differences between the two sites, the discussion is not only speculative but also very partial and oriented in the presented conclusions.

AC52: In this study, we provide data on phytoplankton and zooplankton (ciliate and copepods) abundances from Silcam, Lugol samples and plankton nets. The data presented shows a strong top-down control of the phytoplankton at inshore stations. This is further supported by additional parameters showing high abundances of fecal pellets, as well as high ammonium concentrations. These data support the finding that the low phytoplankton standing stocks are most likely related to a strong predation pressure by copepods and ciliates in the uppermost water column. This finding gives strong indication for a top-down control in this zone as stated in the discussion (lines 531-557). We agree with the reviewer that the top-down control by larger predators (e.g. fish) could also have played a role at both zones. However, we have unfortunately not included any fish into our analysis. So, including this argument in the discussion would certainly be speculative.

RC53: Because of the above-mentioned weaknesses, I could not recommend publication, at least with a clearly major change in results uses and discussion.

Detailed remarks

Title: I am not sure after having read the full manuscript that any results were really exploiting the “patchiness” aspects of the data (except maybe the fact that there is a deep chlorophyl maxima), not to talk about “dissecting” this patchiness. On the same aspect, I was really enthusiast about the title which let me think that in-situ optical sensors were used onboard autonomous platforms (gliders)…. While in fact it is not at all the case. The abstract did not clearly correct this and this is only in the material and methods that I finally understood that it was not the case. Because of those two points, the title is misleading and should be changed or corrected.

AC53. We have changed the title. See comment above (AC46).

RC54: Line 19: adaptative sampling should be understandable by someone that are not familiar with gliders deployments: please explain here.

AC54. We have now added a sentence explaining what adaptive sampling is (line 184, track changes). 

RC55: Line 26: “exposed” exposed to what? (maybe just used inshore/offshore)

AC55: We have changed “exposed” to “offshore”.

RC56: Line27: if sub-surface chlorophyll maxima is correlated with a decrease of abundance of those two large phytoplankton cells, maybe this should indicate that other phytoplankton are responsible for this:

AC57. Other phytoplankton groups were also counted, and similar pattern was observed. See AC49.

RC58: Line 91 (section study area): most of this section present why the study area is a great study site in term of scientific background (reduction of seabird etc…). Therefore, I suggest moving most of this section to introduction.

AC58. We appreciate the reviewer’s suggestion but prefer to keep this part in the study area. We want to do this because the focus in the introduction is to use multiple instrumentation in assessing plankton distributions and SCM. Runde is used as a case of study of the applicability of several approaches for observing plankton distributions in the ocean. 

RC59: Line 95-96: Could you please indicate those two currents on the Fig 1 to guide the reader?

AC59: Yes, they are now in Fig. 1a.

RC60: Line 118: Fig 1 should be called since there.

AC60: Fig. 1 is now called in the text (line 138).

RC61: Figure 1: A and B boxes (used latter) should be indicated here. The line style and line width of those boxes should be darker and larger to be easily spotted (this is not the case here). Consider as well that the color choice for depth is potentially bad: impossible to disentangle depth 225 from depth 600. Additionally, I guess those depth should be rather explained with range of depth (e.g. 200-300; 300-900m), except if the Norwegian seafloor is peculiar in its bathymetry.

AC61: We have now added the labels A and B in the figure 1. The line width is now wider and better spotted. The palette color changed so the range from 300-600 is a darker blue (this can be cleared viewed at Fig 1b, in the deepest part of the fjord).

RC62: In what extend data collected with the boat and with the glider could be compared (notably true since the glider in box A operated in deeper waters (225? 600?) than the close-by sampling stations (125m).

AC62: Note that the platforms used in this study are per definition not gliders (buoyancy-driven) but AUVs (propeller-driven). In box A, the bathymetry shows that both stations and AUVs were in regions where the bottom depth was between 100-300 m). The AUV and the CTD profiles from the boat were operated from surface down to approximately 100 m but not down to the bottom depth (>100 m). The mixed layer depth was <100 m in both areas, so the depth of the bathymetry is irrelevant. 

RC63: Line 143: AUV date deployment? Was it the same AUV deployed two times over short period or two deployed over a longer period?

AC63: The date of deployment in now in the main text. 

RC64: Line 167-171: automatic recognition is often not enough (but see https://doi.org/10.1146/annurev-marine-041921-013023 ). Were the images manually inspected ? Was the efficiency of the image recognition evaluated by any means? The images provided as examples in figure 6 do not allow any understanding on the quality of the process. How much images were acquired? How much classified/ verified by an human expert?

AC 64: We have clarified this in AC48.

RC 65: Line 180: ST is an acquisition protocol not a measurement (line 176), please rephrase.

AC65: “ST measurements” was deleted (line 237, track changes). 

RC66: Line 186: nice explanation, but why not having repeated the same kind of explanations for other parameters bellow. (And no, table 1 does not further demystify those)

AC66: We have now added explanations for each parameter. See lines 242-250 (track changes).

RC67: Line 210: this protocol leads to measure chl a in >0.7µm organisms. Which could potentially explain much difference in between some given parameters (see bad coupling with large phytoplankton reported in abstract). Having this compared with Lugol staining and microscopic counts without explaining what count was operated do not help the reader. What minimal size was considered? Or did only the two target phytoplankton species were counted (which have size >20µm at least)?

AC67: We have now clarified our phytoplankton counting methods. See our comment above (AC 49 and 57). 

RC68: Line 216-219: Ethanol fixation can cause shrinkage of organisms and potentially bias the samples. How this may bias the results or did only a count was done?

AC68: We are aware that ethanol can cause shrinkage of organisms. However, in this study, the focus was on zooplankton abundance estimates and taxonomic identification of major zooplankton groups/species but not on biovolume/biomass estimates. The zooplankton samples fixed with ethanol were in good shape so that it was possible to distinguished between the main zooplankton groups (Calanus, copepods, Cladocera, Euphasiidae, Gastropoda veliger, Hydrozoa) and to get reliable abundance estimates.

RC69: Line 243: why did you pooled those samples? Was the station 5 not acquired? If yes, this should be mentioned since the material and methods. From the fig 1, it seems more logical to pool station 4&5 to compare with the glider, station 3 been right in between the two glider areas. I do understand that this grouping was made on hydrographic structure … which is never shown on a per profile basis.

AC69: We apologize for a slight mistake here. Pooling of the Silcam data was from stations 4 and 5. Pooling from other data (nutrient concentrations, phytoplankton, and zooplankton abundances) was from station 3-5. Grouping was done to highlight the contrasting observations (nutrient concentrations, phytoplankton and zooplankton abundances) based on the distinct hydrographic structure (30m vs 50m) as observed and mentioned by the reviewer. This make the visualization of the contrasting areas much easier to be noticed (the pink versus grey colors in Figs 5-8). The vertical profile from each station is now shown in Fig. 4e,f. 

RC70: Results:

Line 266-267: there is no way to say that the vertical structure of AUV and CTD show similar patterns here, and it is impossible to judge this without plotting those side by side with similar measurements: here temperature, chlorophyl and salinity are shown from AUV while density is displayed only from boat measurements (no temperature/salinity, chl a?).

AC70. The reviewer is correct, so we changed Figure 3 to include temperature and salinity profiles for each station. Note that stations 1 and 2 have the similar structure than box A from the AUV survey and stations 3, 4 and 5 have similar structure to box B (Fig 3d,e).

RC71: Is the difference of MLD due to the inshore/offshore nature of the stations or did the wind mixing occurred somehow between station 2 and 3?

AC71: Strong winds occurred throughout the whole cruise (see Fig 3). The difference in the MLD could be due to exposure of the site to the wind mixing (offshore – more exposed to winds, inshore – more protected from the winds because of the several island around). We clarified this in lines 339 – 347.

RC72: Line 290: add space

AC 72. Added

RC 73: Line 302 and onward (not a comment but important for latter comments): here is related that deeper-mixed waters have less nitrates/ higher (large) phytoplankton (but no SCM) and lower zooplankton. On the contrary shallow mixed waters have higher nutrients, low (large) phytoplankton (but high Chla) and high zooplankton. All this looks like the usual symptoms of an ecosystem under top-down control.

AC73: We agree with the clarifications made by the reviewer. We discuss the role of zooplankton top-down control on phytoplankton standing stocks in the discussion section (lines 531-557). In addition, we wanted to point out that an introduction of nutrients into the upper surface layer can happen due to wind stress thus leading to a bottom up effect during periods when mixing of the water column occurs after a long period of stratification (Fig 3, lines 493-510, track changes). 

RC74: Discussion

Line 410-411: very speculative, from the results obtained, this could also be due to a trophic cascade (top down effect) rather than the bottom up effect proposed. But without temporal observations, this is hard to say. Did the AUV were at sea a long time and could provide such temporal vision?

AC74: The AUV was there for a short time (see now table 1). However, data from continuous measurements from a buoy nearby (Fig. 3) provide evidence that offshore waters were stratified for at least one month prior to the storm which resulted in a well-mixed water column (down to at least 40 m or deeper) during the time of sampling. 

RC75: Line 431: reference 41 format in different.

AC75: Format fixed.

RC76: Line 442-458: this part is really speculative since those observations could be the results of several inhomogeneous measurements: Chla both seen by sensors and measured on filters is usually from small size fraction (filtered onto 0.2µm filter). The fact to have measured low abundance of two large phytoplankton is not an indication that all phytoplankton were low (and higher Kd seem to indicate larger particles load….. usually correlated with small phytoplankton or sediments). All this part relies on the fact that all phytoplankton are supposed to be represented by the two target species identifies and counted on microscope. Since this is very likely not the case, the whole part needs to be modified to take this into account.

Another important thing is that those high chla/ low phytoplankton have large amounts of zooplankton (which prefers larger preys) and may help in amplifying this discrepancy.

AC76: The reviewer missed out the information that other phytoplankton were included. We have now clarified in the text that we included flagellates above 4 µm (see comments AC49). Chlorophyll samples are usually filtered onto GFF filters, which has a fixed mesh size (0.7µm). It is also important to know that this is not the oligotrophic gyre, where picophytoplankton has a large contribution to biomass and fluorescence. Higher Kd could be an indication of high zooplankton and fecal pellets (this is addressed in the manuscript, lines 566-569). We ask for the reviewer to take into consideration that other phytoplankton (not just Tripos and Proboscia) were included. We agree with top-down controls on phytoplankton and that is addressed in lines 531-557.

RC77: Line 472-484: all this explanation (on differences between zooplankton and phytoplankton) relies on the assumption that all zooplankton reacts like ciliates (with rapid respond to phytoplankton bloom). However, half of zooplankton measured are instead copepods, which does have a lifetime of the level of months and then could not respond within the (ungiven) timeframe between mixing and sampling. Therefore, this part sound really speculative. Moreover, it relies on the untold assumption that conditions were about similar before the mixing, which is again unlikely to be the case. One other explanation is that deeply mixed waters stations are naturally low in abundance of zooplankton…. And this could be due to higher predation onto zooplankton (fishes?). Since the described process sounds really in line with top-down control of those ecosystems, I suggest the authors to rethink their results in this respect and not necessarily try to explain those by bottom up control processes (note that both could happens together).

AC77: The reviewer is correct about the ciliate comment, so we removed the ciliate part of the discussion (see lines 531-557). We also removed the part that gives a wrong impression that the areas were similar before. As mentioned earlier, we agree with the reviewer that a strong top-down control was obviously acting in both zones and that predation pressure from larger zooplankton or fish might also have occurred. However, we have no data to proof predation by fish but only data on cilates and mesozooplankton available. In addition to top-down control, bottom-up processes happen in parallel especially at times when wind stress and mixing increases. This can then lead to an introduction of nutrients to the upper water column and stimulate phytoplankton production. The buoy data provided indication that the water column in the area was stratified for at least one month before the wind surge. Therefore, we state that this wind stress could have introduced nutrients to the upper water column and boosted phytoplankton abundances in offshore, well-mixed waters (we see a drawdown down to 60m), thus leading potentially to a bottom-up effect. We also think that top-down effect is occurring in inshore areas due to the high copepods, ammonium and fecal pellet concentrations and low phytoplankton concentrations, giving a strong argument of a top-down control as previously brought up by the reviewer. 

RC78: Figure 5: the bottom panel (e) have a depth axis together with taxonomy

AC78: Fixed.

RC79: Figure 6: this figure has really a bad quality and do not allow to neither read correctly the axis not see the images that are supposed to represent plankton species … is the present state, this is not possible at all and could not be used at all to understand results. Such low quality figure (not that all the other ones were badly pixeled too) is unacceptable and could not be reviewed.

AC79: We have now included Fig 2, which shows the collage of the plankton with the two magnification lenses at a higher resolution. The plankton images in 6a and Fig 6b is now included in the supplementary material as a full page and has a higher resolution, so the quality of the image is now improved.

---

## [Editor Report · Decision Letter 1]

26 Jul 2022

PONE-D-22-12849R1Contrasting phytoplankton-zooplankton distributions observed through autonomous platforms, in-situ optical sensors and discrete sampling.PLOS ONE

Dear Dr. Fragoso,

Thank you for submitting your manuscript to PLOS ONE. After careful consideration, we feel that it has merit but does not fully meet PLOS ONE’s publication criteria as it currently stands. Therefore, we invite you to submit a revised version of the manuscript that addresses the points raised during the review process.

We look forward to receiving your revised manuscript.

Kind regards,

Emmanuel S. Boss

Academic Editor

PLOS ONE

Journal Requirements:

Additional Editor Comments:

Dear authors,

I have read the reviewers comments and your answers as well as the revised manuscript and found your answer and the manuscript adequate.

One minor point you may want to revise is the interpretation of the data as suggesting either top-down control or bottom up control in each of the two cases.

I think that looking at the ocean as usually being close to steady state (loss=growth) with perturbation in the system allowing for improved growth (increased biomass) or with stagnation resulting in reduced growth (and hence decreasing biomass) with mixotrophic organisms contributing to tip further the balance may be a useful framework.

Thus when NPP is strong grazing maybe equally strong resulting in no net phytoplankton biomass accumulation but high concentration of grazing indicators. Similarly, when NPP is weak, there can be no net phytoplankton biomass accumulation, but low concentration of grazing indicators.

It is very rare (e.g. harmful algal blooms) where only one side is winning (hence observed accumulation of phytoplankton in the field are typically an order of magnitude smaller than their specific growth rates).

Feel free to not revise based on this advice.

Best, Emmanuel

---

## [Author Response · Author response to Decision Letter 1]

10 Aug 2022

Dear Professor Boss:

I, with my co-authors, would like to thank you for the opportunity to resubmit our article now entitled “Contrasting phytoplankton-zooplankton distributions observed through autonomous platforms, in-situ optical sensors and discrete sampling.” for publication in Plos One. We would like to thank you for your suggestion. We agree that interpretation of these data should not solely be restrained in bottom-up versus top-down in both conditions. We certainly agree that phytoplankton and zooplankton concentrations in this study are tightly coupled with herbivore-phytoplankton interactions, particularly in June, when strong top-down pressure usually occurs. Yet, we cannot rule out the potential effect of episodic storms in providing nutrients for the phytoplankton. To make sure we consider both processes in this paper, we added few sentences referring that the effect of storm might have imbalanced the predator-prey relationship, rather than solely boosting phytoplankton division rates as previously suggested (lines 482, 497, 584, track changes). We have included a new citation from Behrenfeld and Boss (2018) in these lines suggesting the influence of physical perturbations (deep mixing) in slowing down herbivore-phytoplankton interactions. We also added a citation from Castellani and Edwards (2017) for zooplankton identification in the methods section (lines 274). Changes are in the “track changes” version.

Thank you for your consideration, and we hope the paper can now be accepted.

Sincerely,

Glaucia Fragoso

---

## [Editor Report · Decision Letter 2]

17 Aug 2022

Contrasting phytoplankton-zooplankton distributions observed through autonomous platforms, in-situ optical sensors and discrete sampling.

PONE-D-22-12849R2

Dear Dr. Fragoso,

We’re pleased to inform you that your manuscript has been judged scientifically suitable for publication and will be formally accepted for publication once it meets all outstanding technical requirements.

Kind regards,

Emmanuel S. Boss

Academic Editor

PLOS ONE

Additional Editor Comments (optional):

Dear authors,

I am happy to accept your important contribution to PLOS One. 

Also, thank you for taking my non-binding comments into heart.

I was NOT fishing for citations, but if you found the BB2018 paper of relevance that is great.

One more comment: if the growth rate of phytoplankton gets a boost (due to nutrient input), it will likely, in a first stage cause growth processes > loss processes (resulting in accumulation, or dP/dt>0).

This is likely to be followed by increase in losses (due to increased contact rate) bringing things back to quasi steady-state (growth=loss).

Hence to keep accumulating over time (long term 'blooming') growth needs to continuously improve relative to loss. This does not mean that loss is not decreased during mixing (it might very well do so) but simply that what we observed is the imbalance between growth and loss processes, never one in the absence of the other (except, maybe during HABs).

All the best, Emmanuel
---

## [Editor Report · Acceptance letter]

26 Aug 2022

PONE-D-22-12849R2 

Contrasting phytoplankton-zooplankton distributions observed through autonomous platforms, *in-situ* optical sensors and discrete sampling. 

Dear Dr. Fragoso:

I'm pleased to inform you that your manuscript has been deemed suitable for publication in PLOS ONE. Congratulations! Your manuscript is now with our production department. 

Kind regards, 

on behalf of

Dr. Emmanuel S. Boss 

Academic Editor

PLOS ONE